# Quantum Algorithms for Projection-Free Sparse Convex Optimization

## Abstract

This paper considers the projection-free sparse convex optimization problem for the vector domain and the matrix domain, which covers a large number of important applications in machine learning and data science. For the vector domain $\mathcal{D} \subset \mathbb{R}^d$, we propose two quantum algorithms for sparse constraints that finds a $\varepsilon$-optimal solution with the query complexity of $O(\sqrt{d}/\varepsilon)$ and $O(1/\varepsilon)$ by using the function value oracle, reducing a factor of $O(\sqrt{d})$ and $O(d)$ over the best classical algorithm, respectively, where $d$ is the dimension. For the matrix domain $\mathcal{D} \subset \mathbb{R}^{d \times d}$, we propose two quantum algorithms for nuclear norm constraints that improve the time complexity to $\tilde{O}(rd/\varepsilon^2)$ and $\tilde{O}(\sqrt{r}d/\varepsilon^3)$ for computing the update step, reducing at least a factor of $O(\sqrt{d})$ over the best classical algorithm, where $r$ is the rank of the gradient matrix. Our algorithms show quantum advantages in projection-free sparse convex optimization problems as they outperform the optimal classical methods in dependence on the dimension $d$.

## 1 Introduction

In this paper, we consider the following *constrained* optimization problem of the form

$$\min_{\boldsymbol{x} \in \mathcal{D}} f(\boldsymbol{x}), \tag{1}$$

such objective covers many important application in operations research and machine learning. We are interested in the case where 1) the objective function $f$ is convex and continuously differentiable, and 2) the domain $\mathcal{D} \subset \mathbb{R}^d$ is a feasible set that is convex, and the dimension $d$ is high. Typical instances of such high-dimensional optimization problems include multiclass classification, multitask learning, matrix learning, network systems and many more Garber & Hazan (2016); Hazan & Kale (2012); Hazan et al. (2012); Jaggi (2013); Dudik et al. (2012); Zhang et al. (2012); Harchaoui et al. (2015); Hazan & Luo (2016). As an example, for matrix completion, the optimization problem is:

$$\min_{X \in \mathbb{R}^{m \times n}, \|X\|_{\mathrm{tr}} \leq k} \sum_{(i,j) \in \Omega} (X_{i,j} - Y_{i,j})^2, \tag{2}$$

where $X$ is the matrix to be recovered, $\Omega$ denotes the observed elements, $Y_{i,j}$ is the observed known value at position $(i, j)$, and $\|X\|_{\mathrm{tr}} \leq k$ represents the trace norm (nuclear norm) constraint.

Compared with unconstrained convex optimization problems, optimizing Equation (1) involves handling constraints, which introduces new challenges. A straightforward method for optimizing Equation (1) is the projected gradient descent approach Levitin & Polyak (1966). This method first takes a step in the gradient direction and then performs the projection to satisfy the constraint. However, in practice, the dimensions of the feasible set can be very large, leading to prohibitively high computational complexity. For example, when solving Equation (2), the projection step involves performing a singular value decomposition (SVD), whose time complexity is $O(mn \min\{m, n\})$ ($O(d^3)$ for $X \in \mathbb{R}^{d \times d}$). Compared to the projected gradient descent approach, the Frank-Wolfe (FW) method (also known as the conditional gradient method) is more efficient when dealing with structured constrained optimization problems. Rather than performing projections, it solves a computationally efficient linear sub-problem to ensure that the solution lies within the feasible set $\mathcal{D}$. When solving Equation (2), the time complexity of the Frank-Wolfe method is $O(mn)$ ($O(d^2)$ for $X \in \mathbb{R}^{d \times d}$), which is significantly lower than the complexity of SVD-based projections. Since the Frank-Wolfe

method is efficient for optimizing many difficult machine learning problems, such as low-rank constrained problems and sparsity-inducing constrained problems, it has attracted significant attention and has been applied to solving Equation (3) and many of its variants.

Despite the efficiency of FW in handling structured constraints, it still incurs significant computational overhead when dealing with high-dimensional problems. The bottleneck of the computation is the linear subproblem over $\mathcal{D}$, which is either assumed to have efficient implementation or simply follows existing classical oracles, such as Dunn & Harshbarger (1978); Jaggi (2013); Garber & Hazan (2016). The overhead of these oracles, however, grows linearly or superlinearly in terms of dimension $d$.

Recently, quantum computing has emerged as a promising new paradigm to accelerate a large number of important optimization problems (see Appendix A.4). We aim to take a thorough investigation on whether quantum computing can accelerate FW algorithms, in particular the linear sub-problem over structured constraints regarding dimension $d$. We aim to answer the following question:

*Can one utilize quantum techniques to accelerate Frank-Wolfe algorithms in terms of dimension $d$?*

Chen et al. gave an initial answer to this question Chen & de Wolf (2023). They considered the linear regression problem with explicit functional form where the closed form of gradient is provided. Given the precomputed matrix factors of the closed-form objective function stored in specific data structures, they leveraged HHL-based algorithms to accelerate matrix multiplications in calculating the closed-form gradient, leading to a upper bound of $O\left(\sqrt{d}/\varepsilon^2\right)$. In this work, we consider a more general problem where the objective function is a smooth convex function accessible only through a function value oracle, and then we consider a more general constraint conditions (*the latent group norm ball*) to enhance the theoretical framework's applicability. Besides, we also consider the case of matrix feasible set, under different assumptions. To our best knowledge, we are the first one to consider accelerating the matrix case of the FW algorithm by quantum computing.

**Contributions.** We give a systematic study on how to accelerate FW algorithms when $\mathcal{D}$ is either a vector domain $\mathbb{R}^d$, or a matrix domain $\mathbb{R}^{d \times d}$ subject to various structured constraints. Note that our findings can be applied to non-square matrices, we express our results using square matrices for simplicity of presentation (Remark 1). We summarize our contributions as follows.

For the vector domain $\mathcal{D} \subset \mathbb{R}^d$:

- We propose the quantum Frank-Wolfe algorithm for the projection-free sparse convex optimization problem under $\ell_1$ norm constraints (Theorem 1) and the $d$-dimensional simplex $\Delta_d$ (Theorem 2). We achieve a query complexity of $\widetilde{O}(\sqrt{d}/\varepsilon)$ in finding an $\varepsilon$-optimal solution using the function value oracle, reducing a factor of $O(\sqrt{d})$ over the optimal classical algorithm. Furthermore, if the objective function is a Lipschitz continuous function, we prove that the query complexity can be reduced to $O(1/\epsilon)$ by employing the bounded-error Jordan quantum gradient estimation algorithm, at the cost of more qubits and additional gates (Theorem 5). In addition, we consider the generalization to latent group norm constraints (Theorem 6) and achieve a query complexity of $\widetilde{O}\left(\sqrt{|\mathcal{G}|}\|\mathfrak{g}\|_{\max}\right)$, representing an $O\left(\sqrt{|\mathcal{G}|}\right)$ speedup over the classical algorithm. These results are presented in Section 3, Appendix A.1 and A.2. The comparison with the classical methods is shown in Table 1.

- Specifically, we develop a novel quantum subroutine for the Frank-Wolfe linear subproblem over latent group norm constraints, by computing dual norms coherently across all groups in quantum superposition and identifying the dominant group via quantum maximum finding. We establish a novel error propagation analysis for dual norm computation under gradient approximation, deriving bounds via Hölder's inequality that enable precise control of linear subproblem accuracy throughout Frank-Wolfe iterations. The examples in the main text such as the $\ell_1$-norm constrained are special instances of the latent group constraints. In short, we develop quantum subroutines for dominant atom finding and show that the errors can be controlled by setting appropriate parameters.

For the matrix domain $\mathcal{D} \subset \mathbb{R}^{d \times d}$:

- For the projection-free sparse problem under nuclear norm constraints, we propose two complementary quantum Frank-Wolfe algorithms tailored to high-rank and low-rank gradient matrices, respectively (see Appendix A.5). For finding an $\varepsilon$-optimal solution, we achieve a time complexity of $\tilde{O}(rd/\varepsilon^2)$ (Theorem 3) and $\tilde{O}(\sqrt{r}d/\varepsilon^3)$ (Theorem 4) in computing the update direction, representing an at least $O(\sqrt{d})$ speedup over state-of-the-art classical algorithm, where $r$ is the rank of the gradient matrix. These results are presented in Section 4 and the comparison with the classical methods is shown in Table 2.

- Specifically, in the first algorithm, we simplify the top-$k$ singular vectors extraction method Bellante et al. (2022) by utilizing the quantum maximum finding algorithm, which avoids the overheads of repeated sampling to estimate the factor score ratio, and avoids the overheads of searching the threshold value. In the second algorithm, we introduce the quantum power method to extract the top singular vectors, which reduces the dependence on the rank of the gradient matrix, at the cost of higher sensitivity on solution precision.

Wide range of critical applications can be benefited from the acceleration of QFW, including sparse regression (Lasso), sparse signal recovery, matrix completion, boosting algorithms (e.g., AdaBoost), Support Vector Machines, and density estimation Jaggi (2013). Other applications include signal processing (sparsity constraints via $\ell_1$ norm), game theory (zero sum games with simplex) and SDPs (nuclear norm optimization). The proposed top singular vectors extraction techniques also have a potential application for bi-quadratic programming Li et al. (2024). We discuss some of these applications in Appendix A.6

We notice an independent work on the quantum power method Chen et al. (2025a), whose second algorithm shares a conceptual similarity with our second approach: both iteratively apply quantum matrix-vector multiplication. They assume a sparse-query access to the matrix as input, and achieve a complexity of $\widetilde{O}((d\sqrt{s}/\gamma\varepsilon)^{1+o(1)})$, where $s$ is the sparsity, $\gamma$ is the eigenvalue gap, whereas our method relies on the rank of the matrix, instead of the sparsity. In the case of dense full-rank matrix, their algorithm and ours are consistent in terms of dimensional dependence, which provides mutual validation of correctness.

The remainder of this paper is organized as follows. Section 2 introduces the basic concept of constrained optimization and the classical Frank-Wolfe algorithm. Appendix A.3 introduces the notations and assumptions of quantum computing. Section 3 and 4 presents our quantum FW methods for vector domain and matrix domain, respectively. Extension for the vector cases are presented in Appendix A.1 and A.2. Extended related works are presented in Appendix A.4, and we conclude with a discussion about the future work in Section 5. Proof details are given in Appendix B.

Table 1: Classical algorithms V.S. quantum algorithms of the vector case, where $C_f$ is the curvature of the objective function $f$, $\varepsilon$ is the precision of the solution, $d$ is the dimension of the domain, $G$ is the Lipschitz parameter of the objective function, $p$ is the failure probability.

| Optimization Domain | Constraints | Algorithm | Iteration | Query complexity | Qubits | Gates |
|---|---|---|---|---|---|---|
| Sparse Vectors | $\|\cdot\|_1$-ball | FW Jaggi (2013) | $O(C_f/\varepsilon)$ | $O(d)$ | | |
| | | **QFW (Theorem 1)** | $O(C_f/\varepsilon)$ | $O(\sqrt{d}\log(C_f/p\varepsilon))$ | $O\left(d+\log\frac{1}{\varepsilon}\right)$ | $O(\sqrt{d})$ |
| | | **QFW (Theorem 5)** | $O(C_f/\varepsilon)$ | $O(1)$ | $O\left(d\log\frac{Gd}{p\varepsilon}\right)$ | $O(d\log d)$ |
| Sparse non-neg. vectors | Simplex $\Delta_d$ | Frank-Wolfe Jaggi (2013) | $O(C_f/\varepsilon)$ | $O(d)$ | | |
| | | **QFW (Theorem 2)** | $O(C_f/\varepsilon)$ | $O(\sqrt{d}\log(C_f/p\varepsilon))$ | $O\left(d+\log\frac{1}{\varepsilon}\right)$ | $O(\sqrt{d})$ |
| | | **QFW (Theorem 5)** | $O(C_f/\varepsilon)$ | $O(1)$ | $O\left(d\log\frac{C_f Gd}{p\varepsilon}\right)$ | $O(d\log d)$ |
| Latent group sparse vectors | $\|\cdot\|_G$-ball | FW Jaggi (2013) | $O(C_f/\varepsilon)$ | $O(\sum_{g\in\mathcal{G}}|g|)$ | | |
| | | **QFW (Theorem 6)** | $O(C_f/\varepsilon)$ | $O\left(\sqrt{|\mathcal{G}|}|\mathfrak{g}|_{\max}\log(C_f/p\varepsilon)\right)$ | $O(d+\log|\mathcal{G}|+|\mathfrak{g}|_{\max}\log(1/\varepsilon))$ | $\tilde{O}(\sqrt{|\mathcal{G}|}\cdot|\mathfrak{g}|_{\max})$ |

## 2 PRELIMINARIES

### 2.1 NOTATIONS AND ASSUMPTIONS FOR CONSTRAINED OPTIMIZATION PROBLEM

We consider constrained convex optimization problems of the form

$$\min_{\boldsymbol{x}\in\mathcal{D}} f(\boldsymbol{x}), \tag{3}$$

where $\boldsymbol{x}\in\mathbb{R}^d$, $f:\mathbb{R}^d\to\mathbb{R}$, and $\mathcal{D}\subseteq\mathbb{R}^d$ is the constraint set. In addition, as usually in constrained convex optimization, we also make the following assumptions:

Table 2: Classical algorithms V.S. quantum algorithms of the matrix case, where $C_f$ is the curvature of the objective function $f$, $\varepsilon$ is the precision of the solution, $d$ is the dimension of the domain, $T_\nabla$ is the times required to evaluate $\nabla f$; $\sigma_1(M)$ and is the largest and the second largest singular value, respectively; $r$ is the rank of the gradient matrix; $\gamma'_{\min}$ is a factor which depends on the relation of the singular value distribution of the gradient matrix and the direction of the initial vector.

| Domain | Constraints | Algorithm | Iteration | Complexity of the Update Computing |
|---|---|---|---|---|
| Sparse Matrices | $\|\cdot\|_{tr}$-ball | FW with Power Method Jaggi (2013) | $O(C_f/\varepsilon)$ | $O\left(\frac{\sigma_1(M)d^2}{(\sigma_1(M)-\sigma_2(M))\varepsilon} + T_\nabla\right)$ |
| | | FW with Lanczos Method Jaggi (2013) | $O(C_f/\varepsilon)$ | $O\left(\frac{\sqrt{\sigma_1(M)}d^2}{\sqrt{(\sigma_1(M)-\sigma_2(M))\varepsilon}} + T_\nabla\right)$ |
| | | **FW with QTSVE (Theorem 3)** | $O(C_f/\varepsilon)$ | $\tilde{O}\left(\frac{r\sigma_1^3(M)d}{(\sigma_1(M)-\sigma_2(M))\varepsilon^2} + T_\nabla\right)$ |
| | | **FW with QPM (Theorem 4)** | $O(C_f/\varepsilon)$ | $\tilde{O}\left(\frac{\sqrt{r}\sigma_1^4(M)d}{(1-\sigma_1(M))\gamma'^3_{\min}\varepsilon^3} + T_\nabla\right)$ |

---

**Algorithm 1** Classical Frank-Wolfe Algorithm with Approximate Linear Subproblems

---

1: **Input:** Solution precision $\varepsilon$, iterations $T$.
2: **Output:** $\boldsymbol{x}^{(T)}$ such that $f(\boldsymbol{x}^T) - f(\boldsymbol{x}^*) \leq \varepsilon$.
3: **Initialize:** Let $\boldsymbol{x}^{(1)} \in \mathcal{D}$.
4: **for** $t = 1, ..., T$ **do**
5:     Let $\gamma_t = \frac{2}{t+2}$.
6:     Find direction $\boldsymbol{s} \in \mathcal{D}$ such that

$$\langle \boldsymbol{s}, \nabla f(\boldsymbol{x}^{(t)}) \rangle \leq \min_{\hat{\boldsymbol{s}} \in \mathcal{D}} \langle \hat{\boldsymbol{s}}, \nabla f(\boldsymbol{x}^{(t)}) \rangle + \frac{\delta}{2}\gamma_t C_f. \tag{5}$$

7:     Update $\boldsymbol{x}^{(t+1)} = (1 - \gamma_t)\boldsymbol{x}^{(t)} + \gamma_t \boldsymbol{s}$.
8: **end for**

---

**Assumption 1.** *$f$ is convex and $L$-smooth, i.e., the gradient of $f$ satisfies $\|\nabla f(\boldsymbol{x}) - \nabla f(\boldsymbol{y})\|_2 \leq L\|\boldsymbol{x} - \boldsymbol{y}\|_2$ for any $\boldsymbol{x}, \boldsymbol{y} \in \mathbb{R}^d$.*

**Assumption 2.** *$\mathcal{D}$ is compact and convex, and the diameter of $\mathcal{D}$ has an upper bound $D$, i.e., $\forall x, y \in \mathcal{K}, \|x - y\|_2 \leq D$.*

Typically, solving $\operatorname{argmin}_{\boldsymbol{x} \in \mathcal{D}} \boldsymbol{x}^\top \boldsymbol{y}$ for any $\boldsymbol{y} \in \mathbb{R}^d$, is much faster than the projection operation onto $\mathcal{D}$ (i.e., solving $\operatorname{argmin}_{\boldsymbol{x} \in \mathcal{D}} \|\boldsymbol{x} - \boldsymbol{y}\|$). Examples of such domains include the set of sparse vectors, bounded norm matrices, flow polytope and many more Hazan & Kale (2012). Therefore, for such domains, the basic idea of the Frank-Wolfe algorithm is to replace the projection operation with a linear optimization problem.

In the design and analysis of the Frank-Wolfe algorithm, one important quantity is the curvature $C_f$, which measures the "non-linearity" of $f$ and is defined as follows,

$$C_f = \sup_{\boldsymbol{x}, \boldsymbol{s} \in \mathcal{D}, \beta \in [0,1], \boldsymbol{y} = \boldsymbol{x} + \beta(\boldsymbol{s} - \boldsymbol{x})} \frac{2}{\beta^2} \times (f(\boldsymbol{y}) - f(\boldsymbol{x}) - \langle \boldsymbol{y} - \boldsymbol{x}, \nabla f(\boldsymbol{x}) \rangle). \tag{4}$$

By Lemma 7 of Jaggi (2013), the curvature can be bounded as $C_f \leq LD^2$.

## 2.2 CLASSICAL FRANK-WOLFE ALGORITHM

The classical Frank-Wolfe algorithm is given in Algorithm 1. The key step is the linear subproblem of Equation (5) which seeks an approximate minimizer in $\mathcal{D}$ of $\langle \boldsymbol{s}, \nabla f(\boldsymbol{x}^{(t)}) \rangle$. Classically, the per-step cost is $O(N)$ where $N$ is the number of elements that need to be searched which introduces a large $O(N)$ cost. In this work, we will show that $O(\sqrt{N})$ quantum queries to solve this subproblem.

**Lemma 1.** *[Jaggi (2013), Theorem 1] For each $t \geq 1$, the iterates of Algorithm 1 satisfy*

$$f(\boldsymbol{x}^{(t)}) - f(\boldsymbol{x}^*) \leq \frac{2C_f}{t+2}(1 + \delta), \tag{6}$$

*where $x^*$ is the optimal solution to Equation (3), and $\delta$ is the solution quality to which the internal linear subproblems are solved. That is, one can use $O(\frac{(1+\delta)C_f}{\varepsilon})$ iterations to have a $\varepsilon$-opt solution.*

---

**Algorithm 2** Quantum Frank-Wolfe Algorithm for Sparsity/Simplex Constraint

---

1: **Input:** Solution precision $\varepsilon$, gradient precision $\{\sigma_t\}_{t=1}^T$.
2: **Output:** $\boldsymbol{x}^{(T)}$ such that $f(\boldsymbol{x}^T) - f(\boldsymbol{x}^*) \leq \varepsilon$.
3: **Initialize:** Let $\boldsymbol{x}^{(1)} \in \mathcal{D}$.
4: Let $T = \frac{4C_f}{\varepsilon} - 2$.
5: **for** $t = 1, ..., T$ **do**
6:     Let $\gamma_t = \frac{2}{t+2}$.
7:     Prepare quantum state $\sum_{i=0}^{d-1} |i\rangle \left| \boldsymbol{x}^{(t)} \right\rangle |0\rangle$.
8:     Perform quantum gradient circuit (Lemma 3) to get $\sum_{i=0}^{d-1} |i\rangle \left| \boldsymbol{x}^{(t)} \right\rangle \left| \frac{f(\boldsymbol{x}^{(t)} + \sigma_t \boldsymbol{e}_i) - f(\boldsymbol{x}^{(t)})}{\sigma_t} \right\rangle$.
9:     Apply quantum maximum finding to the absolute value of the third register (to the third register directly for the simplex constraint, respectively) (Lemma 4), and then measure the first register to obtain measurement result $i_t$.
10:     Set $\boldsymbol{s} = -\boldsymbol{e}_{i_t}$. Update $\boldsymbol{x}^{(t+1)} = (1 - \gamma_t)\boldsymbol{x}^{(t)} + \gamma_t \boldsymbol{s}$.
11: **end for**

---

# 3 QUANTUM FRANK-WOLFE ALGORITHMS OVER VECTORS

## 3.1 QUANTUM FRANK-WOLFE WITH SPARSITY CONSTRAINTS

We first consider the optimization problem

$$\min f(\boldsymbol{x}), \text{ s.t. } \boldsymbol{x} \in \mathbb{R}^d, \|\boldsymbol{x}\| \leq 1, \tag{7}$$

where the sparsity constraint $\mathcal{D} = \{\boldsymbol{x} \in \mathbb{R}^d : \|\boldsymbol{x}\|_1 \leq 1\}$.

By Section 4 of Jaggi (2013), any linear function attains its minimum over a convex hull at a vertex. Thus, for the $\ell_1$ norm problem, the exact minimizer (i.e, corresponding to $\delta = 0$) of Equation (5) is $\hat{\boldsymbol{s}} = -\boldsymbol{e}_{i_t}$ with

$$i_t \in \underset{i \in [d]}{\operatorname{argmax}} |\nabla_i f(\boldsymbol{x}^{(t)})|, \tag{8}$$

i.e., it is a coordinate corresponding to the largest absolute value of the gradient component.

Our approach will be to construct an approximate quantum maximum gradient component finding algorithm to find such an $i_t$.

**Quantum access model $U_f$.** In this subsection, we assume that the value of the loss function is accessed via a function value oracle as shown in Assumption 3.

**Assumption 3.** *There is a unitary $\boldsymbol{U}_f$ that, in time $T_f$, returns the function value, i.e., $\boldsymbol{U}_f : |\boldsymbol{x}\rangle |a\rangle \rightarrow |\boldsymbol{x}\rangle |a + f(\boldsymbol{x})\rangle$, for any $a$, where $|\boldsymbol{x}\rangle := |x_1\rangle |x_2\rangle ... |x_d\rangle$.*

The preparation of the input state in Step 7 of Algorithm 2 is efficient. Initialize the algorithm at $\boldsymbol{x}^{(0)} = 0$, each Frank-Wolfe step adds a single coordinate direction to the solution. Specifically, the update rule $\boldsymbol{x}^{(t+1)} = (1 - \gamma_t)\boldsymbol{x}^{(t)} + \gamma_t \boldsymbol{s}_t$—where $\boldsymbol{s}_t$ is a standard basis vector—implies that the solution $\boldsymbol{x}^{(t)}$ after $t$ iterations is a sparse vector with at most $t$ non-zero components. Consequently, the quantum state $\left| x^{(t)} \right\rangle$ is a sparse computational basis state. This state can be perform an incremental update, setting at most one new coordinate to a non-zero value per iteration. The gate complexity for this sparse update is $O(t)$. The total number of iterations $T$ required for an $\varepsilon$-optimal solution is $O(1/\varepsilon)$, which is independent of the dimension $d$. Therefore, the state preparation overhead per iteration remains $O(1/\varepsilon)$, completely decoupled from the potentially large dimension $d$.

**Quantum gradient circuit.** Next, we present a general unitary $U_g$ to approximate the gradient $\nabla f(\boldsymbol{x}_t)$. Specifically, we use the forward difference $g_i(\boldsymbol{x}_t) = \frac{f(\boldsymbol{x}_t + \sigma \boldsymbol{e}_i) - f(\boldsymbol{x}_t)}{\sigma}$ to approximate each item of $\nabla_i f(\boldsymbol{x}_t)$ with $\ell_\infty$ error $\varepsilon_g$, i.e., $\|\nabla f(\boldsymbol{x}_t) - g(\boldsymbol{x}_t)\|_\infty \leq \varepsilon_g$, where $\sigma$ is the tunable parameter for the desired accuracy.

**Lemma 2** (Theorem 3.1 Berahas et al. (2022)). *Under* **Assumption 1**, *let* $g_i(\boldsymbol{x}) = \frac{f(\boldsymbol{x}+\sigma e_i)-f(\boldsymbol{x})}{\sigma}$, *then for all* $\boldsymbol{x} \in \mathbb{R}^d$,

$$\|g(\boldsymbol{x}) - \nabla f(\boldsymbol{x})\|_2 \leq \frac{\sqrt{d}L\sigma}{2}. \tag{9}$$

**Lemma 3.** *Given access to the quantum function value oracle* $\boldsymbol{U}_f$, *there exists a quantum circuit to construct a quantum error bounded gradient oracle* $\boldsymbol{U}_g : |i\rangle |\boldsymbol{x}\rangle |0\rangle \rightarrow |i\rangle |\boldsymbol{x}\rangle |g_i(\boldsymbol{x})\rangle$, *where* $g_i(\boldsymbol{x}) = \frac{f(\boldsymbol{x}+\sigma e_i)-f(\boldsymbol{x})}{\sigma}$ *is the* $i$-*th component of the gradient and* $\sigma$ *is the tunable parameter, with two queries to the quantum function value oracle.*

The proof is given in Appendix B.1.

**Quantum maximum finding circuit.** Based on $\boldsymbol{U}_g$, leveraging the quantum minimum-finding algorithm Durr & Hoyer (1996), we give an approximate search of the maximum gradient component as shown in Lemma 4, with proof given in Appendix B.2. Note that Algorithm 3 in the matrix section of this work also utilizes quantum maximum finding, but with a non-uniform input state. We also provide a proof in Appendix B.2 that the quantum maximum finding procedure is applicable to non-uniform input states.

**Lemma 4.** *(Approximate maximum gradient component finding) Given access to the quantum error bounded gradient oracle* $\boldsymbol{U}_g : |i\rangle |\boldsymbol{x}\rangle |0\rangle \rightarrow |i\rangle |\boldsymbol{x}\rangle |g_i(\boldsymbol{x})\rangle$ *s.t. for each* $i \in [d]$, *after measuring* $|g_i(\boldsymbol{x})\rangle$, *the measured outcome* $g_i(\boldsymbol{x})$ *satisfies* $|g_i(\boldsymbol{x}) - \nabla f_i(\boldsymbol{x})| \leq \epsilon$. *There exists a quantum circuit* $\mathcal{A}_{\max}$ *that finds the index* $i^*$ *that satisfies* $\nabla f_{i^*}(\boldsymbol{x}) \geq \max_{j\in[d]} \nabla f_j(\boldsymbol{x}) - 2\epsilon$ *or* $|\nabla f_{i^*}(\boldsymbol{x})| \geq \max_{j\in[d]} |\nabla f_j(\boldsymbol{x})| - 2\epsilon$, *using* $O(\sqrt{d}\log(\frac{1}{\delta}))$ *applications of* $\boldsymbol{U}_g$, $\boldsymbol{U}_g^\dagger$ *and* $O(\sqrt{d})$ *elementary gates, with probability* $1 - \delta$. *For the non-uniform initial state, let* $p$ *be the initial measurement probability of the maximum component, then the algorithm finds the maximum with query complexity of* $O(\frac{1}{\sqrt{p}}\log(\frac{1}{\delta}))$.

**Convergence Analysis.** Now we can conduct the convergence analysis with the help of approximate maximum finding sub-routine and show how to choose appropriate parameters, which gives Theorem 1, with proof given in Appendix B.3.

**Theorem 1.** *(Quantum FW over the sparsity constraint) By setting* $\sigma_t = \frac{C_f}{\sqrt{d}L(t+2)}$ *for* $t \in [T]$, *the quantum algorithm (Algorithm 2) solves the sparsity constraint optimization problem for any precision* $\varepsilon$ *such that* $f(\boldsymbol{x}^T) - f(\boldsymbol{x}^*) \leq \varepsilon$ *in* $T = \frac{4C_f}{\varepsilon} - 2$ *rounds, succeed with probability* $1 - p$, *with* $O\left(\sqrt{d}\log\frac{C_f}{p\varepsilon}\right)$ *calls to the function value oracle* $\boldsymbol{U}_f$ *per round.*

If the objective function is a $G$-Lipschitz continues function (i.e. $|f(\boldsymbol{x}) - f(\boldsymbol{y})| \leqslant G\|\boldsymbol{y} - \boldsymbol{x}\|, \quad \forall \boldsymbol{x}, \boldsymbol{y} \in \mathcal{D}$), an alternative approach for estimating the gradient of the objective function involves employing the bounded-error Jordan algorithm to improve the query complexity of each iteration to $O(1)$, at the cost of additional space complexity and extra gate operations. This result is given in Appendix A.1.

## 3.2 Extensions: Quantum Frank-Wolfe for Atomic Sets

Classically, the Frank-Wolfe algorithm has been shown to be well-suited to atomic sets Jaggi (2013), i.e. where the constraint set is expressed as the convex hull of another (not-necessarily finite) set $\mathcal{A}$: $\mathcal{D} = \mathsf{conv}(\mathcal{A})$ In this case, the Frank-Wolfe update calculation requires a minimization only over $\mathcal{A}$: $\min_{\hat{\boldsymbol{s}}\in\mathcal{A}}\langle\hat{\boldsymbol{s}}, \nabla f(x^{(t)})\rangle$. The optimization over the $\ell_1$ ball as studied above is a special case of this, since $\{\boldsymbol{x} \in \mathbb{R}^d : \|\boldsymbol{x}\|_1\} = \mathsf{conv}\{\pm e_1, \pm e_2, \ldots, \pm e_d\}$. Note also that quantum optimization over the simplex $\Delta_d = \mathsf{conv}\{e_1, \ldots, e_d\}$ can be done by almost exactly the same method as for the $\ell_1$ case, with the only modification to account for the fact that only the unit vectors need to be optimized over, which gives Theorem 2.

**Theorem 2.** *(Quantum FW over the simplex) By setting* $\sigma_t = \frac{C_f}{\sqrt{d}L(t+2)}$ *for* $t \in [T]$, *the quantum algorithm (Algorithm 2) solves the simplex constraint optimization problem for any precision* $\varepsilon$ *such that* $f(\boldsymbol{x}^T) - f(\boldsymbol{x}^*) \leq \varepsilon$ *in* $T = \frac{4C_f}{\varepsilon} - 2$ *rounds, succeed with probability* $1 - p$, *with* $O\left(\sqrt{d}\log\frac{C_f}{p\varepsilon}\right)$ *calls to the function value oracle* $\boldsymbol{U}_f$ *per round.*

Two more extensions for atomic sets are given in Appendix A.2.

# 4 QUANTUM FRANK-WOLFE ALGORITHMS OVER MATRICES

In this section, we consider the matrix version of the constrained optimization problem in Equation (1), specifically,

$$\min f(X), \text{ s.t. } X \in \mathbb{R}^{d \times d}, \|X\|_{\text{tr}} \le 1, \tag{10}$$

where the sparsity constraint is $\mathcal{D} = \{X \in \mathbb{R}^{d \times d} : \|X\|_{\text{tr}} \le 1\}$. For simplicity of presentation, we first focus on square matrices, i.e., $X \in \mathbb{R}^{d \times d}$ (Remark 1).

**Schatten matrix norm.** In contrast to the vector norm $\|\cdot\|$ on $\mathbb{R}^d$, the corresponding Schatten matrix norm $\|X\|$ is defined as $\|(\sigma_1, ..., \sigma_d)\|$, where $\sigma_1, ..., \sigma_d$ are singular values of $X$. It is known that the dual of the Schatten $\ell_p$ norm is the Schatten $\ell_q$ norm with $1/p + 1/q = 1$. The most prominent example is the trace norm $\|\cdot\|_{\text{tr}}$, also referred to as the nuclear norm or Schatten $\ell_1$ norm, defined as the sum of the singular values $\|X\|_{\text{tr}} = \sum_{i=1}^{d} \sigma_i$.

**Linear subproblem solver.** Following the classical Frank-Wolfe iteration framework, we aim to solve the linear optimization subproblem $\min_{S \in \mathcal{D}} \langle S, \nabla f(X_t) \rangle$ where $X_t$ denotes the iterate matrix at step $t$, and $\langle X, Y \rangle = \text{tr } X^\top Y$ represents the Hilbert-Schmidt inner product. For convenience, let $M = \nabla f(X_t)$ in the rest of this section. To solve this subproblem, one can compute the singular value decomposition (SVD) $M = U \text{diag}(\boldsymbol{\sigma}) V^\top$, where $\boldsymbol{\sigma}$ are singular values of $M$ and $U, V \in \mathbb{R}^{d \times d}$ are orthogonal matrices. Since Schatten norms are invariant under orthogonal transformations, the optimal solution $S \in \mathcal{D}$ for the minimization problem $\min_{S \in \mathcal{D}} \langle S, M \rangle$ takes the forms of $S = U \text{diag}(\boldsymbol{s}) V^\top$, where $\langle \boldsymbol{s}, \boldsymbol{\sigma} \rangle = \|\boldsymbol{\sigma}\|_q$ with $\|\boldsymbol{s}\|_p \le 1$ and $1/p + 1/q = 1$. For the nuclear norm (i.e., $\ell_1$ Schatten norm), this reduces to $S = \boldsymbol{u}\boldsymbol{v}^\top$ where $\boldsymbol{u}, \boldsymbol{v}$ are the left and right top singular vectors of $M$, corresponding to its largest singular value $\sigma_1(M)$. Thus, the core computational task is to efficiently approximate the top singular vectors $\boldsymbol{u}, \boldsymbol{v} \in R^d$, ensuring $|\boldsymbol{u}^\top M \boldsymbol{v} - \sigma_1(M)| \le \varepsilon$.

**Power method and Lanczos method.** Compared with the SVD that requires $O(d^3)$ computational cost per iteration to compute all $d$ singular vectors, extracting only the top singular vector is much easier. Specifically, Kuczyński & Woźniakowski (1992) considers two iterative methods: the power method and the Lanczos method. The power method achieves $|\boldsymbol{u}^\top M \boldsymbol{v} - \sigma_1(M)| \le \varepsilon'$ with the worst-case computation complexity of $O\left(\frac{\sigma_1(M)d^2 \ln d}{(\sigma_1(M)-\sigma_2(M))\varepsilon'}\right)$, while the Lanczos method achieves $|\boldsymbol{u}^\top M \boldsymbol{v} - \sigma_1(M)| \le \varepsilon'$ with the worst-case computation complexity of $O\left(\frac{\sqrt{\sigma_1(M)}d^2 \ln d}{\sqrt{(\sigma_1(M)-\sigma_2(M))\varepsilon'}}\right)$, where $\varepsilon'$ is the additive error. Similar to the convergence analysis in Section 3.1, setting $\varepsilon' = O(\varepsilon)$, the complexity of update computing are $O\left(\frac{\sigma_1(M)d^2 \ln d}{(\sigma_1(M)-\sigma_2(M))\varepsilon}\right)$ and $O\left(\frac{\sqrt{\sigma_1(M)}d^2 \ln d}{\sqrt{(\sigma_1(M)-\sigma_2(M))\varepsilon}}\right)$, respectively.

**Quantum enhancement.** In the following, we propose two quantum subroutines to compute the top singular vector: the quantum top singular vector extraction method and the quantum power method. Note that for the matrix case, we could also assume the same function value oracle and naturally employ an improved Jordan's algorithm to achieve a query complexity advantage in gradient estimation. However, in this work, we aimed to further investigate whether quantum algorithms can accelerate the computational complexity of the update step beyond just query counts. Therefore, the analysis focuses on the update direction computation and assumes that the gradient has been pre-computed and stored in the memory (Remark 3), following the classical convention of excluding gradient evaluation time Jaggi (2013).

First, we assume the following gradient access model for matrix data. A detailed description of this data structure can be found in Section 1.A of Kerenidis & Prakash (2020b).

**Assumption 4** (Quantum access to a matrix). *We assume that we have efficient quantum access to the matrix $M \in R^{d \times d}$. That is, there exists a data structure that allows performing the mapping $|i\rangle |0\rangle \to |i\rangle |M_{i,\cdot}\rangle = |i\rangle \frac{1}{\|M_{i,\cdot}\|} \sum_j^d M_{ij} |j\rangle$ for all $i$, and $|0\rangle \to \frac{1}{\|M\|_F} \sum_i^d \|M_{i,\cdot}\| |i\rangle$ in time $\widetilde{O}(1)$.*

### 4.1 QUANTUM FRANK-WOLFE WITH QUANTUM TOP SINGULAR VECTOR EXTRACTION

Leveraging the quantum access defined in Assumption 4, quantum singular value estimation can be performed efficiently.

**Lemma 5** (Singular value estimation (Theorem 3 Bellante et al. (2022)), Kerenidis & Prakash (2020b))**.** *Let there be quantum access to $M \in R^{d \times d}$, with singular value decomposition $M = \sum_i^d \sigma_i \boldsymbol{u}_i \boldsymbol{v}_i^T$. Let $\epsilon > 0$ be a precision parameter. There exists a quantum circuit for performing the mapping $\frac{1}{\|M\|_F} \sum_i^d \sum_j^d M_{ij} |i\rangle |j\rangle |0\rangle \to \frac{1}{\|M\|_F} \sum_i^k \sigma_i |\boldsymbol{u}_i\rangle |\boldsymbol{v}_i\rangle |\overline{\sigma}_i\rangle$ such that $|\sigma_i - \overline{\sigma}_i| \leq \epsilon$ with probability at least $1 - 1/poly(d)$ in time $O\left(\frac{\|M\|_F poly \log d}{\epsilon}\right)$.*

To extract classical singular vectors corresponding to the largest singular value from a quantum state, $\ell_2$ norm quantum state tomography is required.

**Lemma 6** ($\ell_2$ state-vector tomography Kerenidis et al. (2020; 2019d))**.** *Given a unitary mapping $U_x : |0\rangle \to |\boldsymbol{x}\rangle$ in time $T(U_{\boldsymbol{x}})$ and $\delta > 0$, there is an algorithm that produces an estimate $\overline{\boldsymbol{x}} \in R^d$ with $\|\overline{\boldsymbol{x}}\|_2 = 1$ such that $\|\boldsymbol{x} - \overline{\boldsymbol{x}}\|_2 \leq \delta$ with probability at least $1 - 1/poly(d)$ in time $O\left(T(U_{\boldsymbol{x}})\frac{d \log d}{\delta^2}\right)$.*

**Quantum top singular vector extraction (QTSVE).** The goal of the quantum subroutine in each iteration is to find the top right / left singular vectors of the gradient matrix. First, we prepare the gradient matrix state using the quantum access as stated in Assumption 4, then we perform QSVE to this state. The quantum maximum finding is applied to obtain the quantum state corresponding to the largest singular value. Prepare sufficient quantum states corresponding to the largest singular value until satisfying the requirement of tomography, then perform quantum state tomography to extract the corresponding right / left classical singular vectors. This procedure is shown in Lemma 7, with the proof given in Appendix B.8. Note that the success probability of QTSVE can be improved by repeating it logarithmic times and then taking the average.

**Lemma 7.** *(Quantum top singular vector extraction) Let there be efficient quantum access to a matrix $M \in R^{d \times d}$, with singular value decomposition $M = \sum_i^d \sigma_i \boldsymbol{u}_i \boldsymbol{v}_i^T$. Define $p = \frac{\sigma_1^2(M)}{\sum_{i=1}^d \sigma_i^2}$.*

*There exist quantum algorithms that with time complexity $O\left(\frac{\|M\|_F d poly \log d}{\sqrt{p} \epsilon \delta^2}\right)$, give the estimated top singular value $\overline{\sigma}_1$ of $M$ to precision $\epsilon$ and the corresponding unit estimated singular vectors $\boldsymbol{u}, \boldsymbol{v}$ to precision $\delta$ such that $\|\boldsymbol{u} - \boldsymbol{u}_{top}\| \leq \delta$, $\|\boldsymbol{v} - \boldsymbol{v}_{top}\| \leq \delta$ with probability at least $1 - 1/poly(d)$.*

**Convergence Analysis.** Our quantum Frank-Wolfe algorithm for nuclear norm constraint (Algorithm 3) then follows, with the analysis given in Appendix B.9.

**Theorem 3.** *(Quantum FW with QTSVE) By setting $\delta_t = \frac{C_f}{2(t+2)\sigma_1(M_t)}$ and $\epsilon_t \leq (\sigma_1(M_t) - \sigma_2(M_t))/2$ for $t \in [T]$, the quantum algorithm (Algorithm 3) solves the nuclear norm constraint optimization problem for any precision $\varepsilon$ such that $f(X^T) - f(X^*) \leq \varepsilon$ in $T = \frac{4C_f}{\varepsilon} - 2$ rounds, with time complexity $\tilde{O}\left(\frac{r\sigma_1^3(M_t)d}{(\sigma_1(M_t) - \sigma_2(M_t))\varepsilon^2}\right)$ for computing the update direction per round, where $r$ is the rank of the gradient matrix.*

In computing the update direction, Algorithm 3 reduces a $O(d\varepsilon/r\sigma_1^2(M))$ factor to the power method and $O(d\varepsilon^{1.5}/r\sigma_1^{2.5}(M))$ to the Lanczos method, respectively. See Remark 2 for more information about parameter choosing.

### 4.2 QUANTUM FRANK-WOLFE WITH QUANTUM POWER METHOD

The second framework is to accelerate the power method directly with quantum matrix-vector multiplication method and quantum tomography. The classical power method constructs a sequence $\boldsymbol{z}_0, ..., \boldsymbol{z}_k$, where $\boldsymbol{z}_0 = \boldsymbol{b}$ is drawn uniformly random over a unit sphere $\boldsymbol{b} : \|\boldsymbol{b}\|_2 = 1$, and $\boldsymbol{z}_{i+1} = M^\top M \boldsymbol{z}_i$ for $i \geq 1$, ($\boldsymbol{z}_{i+1} = M M^\top \boldsymbol{z}_i$ for the left singular vector, respectively). After $k = \frac{C_0 \sigma_1(M) \ln d}{\varepsilon}$, we have $\left|\frac{\boldsymbol{z}_k^\top M \boldsymbol{z}_k}{\|\boldsymbol{z}_k\|_2^2} - \sigma_1(M)\right| \leq \varepsilon$, where $C_0$ is a constant.

**Quantum power method (QPM).** Using the quantum access given in Assumption 4, the quantum matrix-vector multiplication can be performed efficiently:

---

**Algorithm 3** Quantum Frank-Wolfe Algorithm for Nuclear Norm Constraint with QTSVE

---

1: **Input:** Solution precision $\varepsilon$, singular value precision $\{\epsilon_t\}_{t=1}^T$, tomography precision $\{\delta_t\}_{t=1}^T$.
2: **Output:** $X^{(T)}$ such that $f(X^T) - f(X^*) \leq \varepsilon$.
3: **Initialize:** Let $X^{(1)} \in \mathcal{D}$.
4: Let $T = \frac{4C_f}{\varepsilon} - 2$.
5: **for** $t = 1, ..., T$ **do**
6:      Let $\gamma_t = \frac{2}{t+2}$.
7:      Prepare $\frac{1}{\|M\|_F} \sum_i^d \sum_j^d M_{ij} |i\rangle |j\rangle |0\rangle$.
8:      Perform QSVE (Lemma 5) to get $\frac{1}{\sqrt{\sum_i^r \sigma_i^2}} \sum_i^r \sigma_i |\boldsymbol{u}_i\rangle |\boldsymbol{v}_i\rangle |\overline{\sigma}_i\rangle, where |\sigma_i - \overline{\sigma}_i| \leq \epsilon_t$.
9:      Apply quantum maximum finding (Lemma 4) to the third register to get $|\boldsymbol{u}_{top}\rangle |\boldsymbol{v}_{top}\rangle |\overline{\sigma}_1\rangle$.
10:      Perform $\ell_2$-norm tomography (Lemma 6), to obtain $\boldsymbol{u}, \boldsymbol{v}$, where $\|\boldsymbol{u} - \boldsymbol{u}_{top}\| \leq \delta_t$, $\|\boldsymbol{v} - \boldsymbol{v}_{top}\| \leq \delta_t$.
11:      Set $S = \boldsymbol{u}\boldsymbol{v}^\top$. Update $X^{(t+1)} = (1-\gamma_t)X^{(t)} + \gamma_t S$.
12: **end for**

---

**Algorithm 4** Quantum Frank-Wolfe Algorithm for Nuclear Norm Constraint with QPM

---

1: **Input:** Solution precision $\varepsilon$, multiplication times $\{k_t\}_{t=1}^T$, multiplication precision $\{\delta_t\}_{t=1}^T$, tomography precision $\{\delta_t'\}_{t=1}^T$.
2: **Output:** $X^{(T)}$ such that $f(X^T) - f(X^*) \leq \varepsilon$.
3: **Initialize:** Let $X^{(1)} \in \mathcal{D}$.
4: Let $T = \frac{4C_f}{\varepsilon} - 2$.
5: **for** $t = 1, ..., T$ **do**
6:      Let $\gamma_t = \frac{2}{t+2}$.
7:      Prepare $\frac{1}{\|M\|_F} \sum_i^d \sum_j^d M_{ij} |i\rangle |j\rangle |\boldsymbol{b}\rangle |\boldsymbol{b}\rangle$, where $\boldsymbol{b}$ is the uniform superposition state.
8:      Apply quantum power method (Lemma 9) to get $\frac{1}{\|M\|_F} \sum_i^d \sum_j^d M_{ij} |i\rangle |j\rangle |\overline{\boldsymbol{z}}_u\rangle |\overline{\boldsymbol{z}}_v\rangle$, where $\|\overline{\boldsymbol{z}}_u - (MM^\top)^k \boldsymbol{b}\| \leq \delta_t, \|\overline{\boldsymbol{z}}_v - (M^\top M)^k \boldsymbol{b}\| \leq \delta_t$.
9:      Perform $\ell_2$-norm tomography (Lemma 6) to obtain $\boldsymbol{u}, \boldsymbol{v}$, where $\|\boldsymbol{u} - \overline{\boldsymbol{z}}_u\| \leq \delta_t', \|\boldsymbol{v} - \overline{\boldsymbol{z}}_v\| \leq \delta_t'$.
10:      Set $S = \boldsymbol{u}\boldsymbol{v}^\top$. Update $X^{(t+1)} = (1-\gamma_t)X^{(t)} + \gamma_t S$.
11: **end for**

---

**Lemma 8.** *(Quantum matrix-vector multiplication (Theorem 4 Bellante et al. (2022)), Chakraborty et al. (2019)) Let there be quantum access to the matrix $M \in R^{d \times d}$ with $\sigma_{max} \leq 1$, and to a vector $\boldsymbol{z} \in R^d$. Let $\|M\boldsymbol{z}\| \geq \gamma'$. There exists a quantum algorithm that creates a state $|\boldsymbol{y}\rangle$ such that $\||\boldsymbol{y}\rangle - |M\boldsymbol{z}\rangle\| \leq \epsilon$ in time $\tilde{O}\left(\frac{1}{\gamma'}\|M\|_F \log(1/\epsilon)\right)$, with probability at least $1 - 1/poly(d)$.*

Apply $2k$ times of quantum matrix-vector multiplication, we can get a quantum state corresponding to $\boldsymbol{z}_k$, as shown in Lemma 9, with proof given in Appendix B.10. A similar process can be constructed to compute $(MM^\top)^k \boldsymbol{b}$ (corresponding to the left singular vector) simultaneously.

**Lemma 9.** *(Quantum power method) Let there be quantum access to the matrix $M \in R^{d \times d}$ with $\sigma_{max} \leq 1$, and to a vector $\boldsymbol{z} \in R^d$. Let $\gamma'_{\min}$ be the lower bound of $\|(M^\top M)^i \boldsymbol{z})\|$ for all $i \in [k]$. There exists a quantum algorithm that creates a state $|\boldsymbol{y}\rangle$ such that $\||\boldsymbol{y}\rangle - |(M^\top M)^k \boldsymbol{z}\rangle\| \leq \delta$ in time $\tilde{O}(\frac{k}{\gamma'_{\min}}\|M\|_F \log(1/\delta))$, with probability at least $1 - O(k/poly(d))$.*

**Convergence Analysis.** After quantum state tomography, we can extract the classical top singular vectors. Note that the success probability of QPM and tomography can be improved by repeating the whole procedure logarithmic times and then taking the average. Our quantum Frank-Wolfe algorithm (Algorithm 4) for nuclear norm constraint then follows, and the parameters choosing and convergence analysis are given in Theorem 4, with the proof given in Appendix B.11.

**Theorem 4.** *(Quantum FW with QPM) By setting $k_t = \frac{2C_0\sigma_1(M_t)\ln d}{\varepsilon}, \delta_t = \delta_t' = \frac{\varepsilon\gamma'_{\min}}{16\sigma_1(M_t)}$ for $t \in [T]$, the quantum algorithm (Algorithm 4) solves the nuclear norm constraint optimization problem for any precision $\varepsilon$ such that $f(X^T) - f(X^*) \leq \varepsilon$ in $T = \frac{4C_f}{\varepsilon} - 2$ rounds, with time*

*complexity $\tilde{O}\left(\frac{\sqrt{r}\sigma_1^4(M_t)d}{(1-\sigma_1(M_t))\gamma'^3_{\min}\varepsilon^3}\right)$ for computing the update direction per round, where $r$ is the rank of the gradient matrix, $C_0$ is a constant and $\gamma'_{\min}$ is the lower bound of $\left\|(M_t^\top M_t)^i \boldsymbol{b}\right\|$ for all $i \in [k]$.*

In computing the update direction, Algorithm 4 reduces a $O(d\varepsilon^2\gamma'^3_{\min}/\sqrt{r}\sigma_1^3(M))$ factor to the power method and $O(d\varepsilon^{2.5}\gamma'^3_{\min}/\sqrt{r}\sigma_1^{3.5}(M))$ to the Lanczos method. A discussion of this section is given in Appendix A.5.

## 5 CONCLUSION AND FUTURE WORK

This paper addresses the projection-free sparse convex optimization problem. We propose several quantum Frank-Wolfe algorithms for both vector and matrix domains, demonstrating the quantum speedup over the classical methods with respect to the dimension of the feasible set.

For future work, we aim to extend quantum Frank-Wolfe methods to stochastic and online optimization frameworks, to characterize quantum advantages in projection-free regret minimization. Meanwhile, Jaggi (2013) highlights several interesting cases involving matrix norms, where classical approaches often rely on computationally expensive singular value decomposition. A potential avenue of interest is determining whether quantum computing can yield greater speedups in such settings. Moreover, as mentioned in Appendix A.5, the gradient in the matrix completion is sparse, which might allow for further acceleration via quantum sparse matrix multiplication, constituting an interesting direction for future research. These investigations would collectively advance the understanding of quantum-enhanced projection-free optimization.

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

**Appendix**

# A    EXTENSION AND DISCUSSION

## A.1    QUANTUM FRANK-WOLFE OVER VECTORS WITH BOUNDED-ERROR JORDAN ALGORITHM

The quantum Frank-Wolfe Algorithm with Bounded-error Jordan's Algorithm is shown in Algorithm 5. We reformulate the results of the bounded-error Jordan algorithm from He et al. (2024) in terms of infinity norm error, with the proof detailed in the Appendix B.4.

---
**Algorithm 5** Quantum Frank-Wolfe Algorithm with Bounded-error Jordan Algorithm
---
1: **Input:** Solution precision $\varepsilon$, gradient precision $\{\sigma_t\}_{t=1}^T$.
2: **Output:** $\boldsymbol{x}^{(T)}$ such that $f(\boldsymbol{x}^T) - f(\boldsymbol{x}^*) \leq \varepsilon$.
3: **Initialize:** Let $\boldsymbol{x}^{(1)} \in \mathcal{D}$.
4: Let $T = \frac{4C_f}{\varepsilon} - 2$.
5: **for** $t = 1, ..., T$ **do**
6:     Let $\gamma_t = \frac{2}{t+2}$.
7:     Using Algorithm 7 to get the whole vector of estimated gradient $\widetilde{\nabla} f_t(\boldsymbol{x}_t)$.
8:     Scan all the component of $\widetilde{\nabla} f_t(\boldsymbol{x}_t)$ to find the coordinate $i_t$ corresponding to the largest absolute value of the estimated gradient component.
9:     Set $\boldsymbol{s} = -\boldsymbol{e}_{i_t}$. Update $\boldsymbol{x}^{(t+1)} = (1 - \gamma_t)\boldsymbol{x}^{(t)} + \gamma_t \boldsymbol{s}$.
10: **end for**
---

**Lemma 10.** *(Lemma 1 He et al. (2024)) If $f$ is $G$-Lipschitz continues and $L$-smooth convex function and can be accessed by a quantum function value oracle, then there exists an quantum algorithm that for any $r > 0$ and $1 \geq \rho > 0$, gives the estimated gradient $g(x)$, which satisfies*

$$\Pr\left[\|g(x) - \nabla f(x)\|_\infty > 8\pi n^2 (n/\rho + 1)Lr/\rho\right] < \rho, \tag{11}$$

*using $O(1)$ applications of $\boldsymbol{U}_f$ and $O(d \log d)$ elementary gates. The space complexity is $O\left(d \log \frac{G\rho}{4\pi d^2 Lr}\right)$.*

The next step is to determine the quantum gradient estimated parameters $r_t$ in each Frank-Wolfe iteration through convergence analysis.

**Theorem 5.** *(Quantum FW with bounded-error Jordan algorithm) By setting $r_t = \frac{\rho C_f}{16\pi d^2 (d/\rho+1)L(t+2)}$ for $t \in [T]$, the quantum algorithm (Algorithm 5) solves the sparsity constraint optimization problem for any precision $\varepsilon$ such that $f(\boldsymbol{x}^T) - f(\boldsymbol{x}^*) \leq \varepsilon$ in $T = \frac{4C_f}{\varepsilon} - 2$ rounds, with $O(1)$ calls to the function value oracle $\boldsymbol{U}_f$ per round.*

The proof is given in Appendix B.5. Substituting the parameter $r_t$ into the space complexity yields the qubit requirement as $O\left(d \log \frac{Gd}{\rho\varepsilon}\right)$. Since each gradient estimation succeeds with probability $1 - \rho$, the probability that all $T$ iterations succeed is at least $1 - T\rho$. By setting $\rho = p/T$, we ensure an overall success probability of at least $1 - p$.

## A.2    MORE EXTENSIONS OVER VECTORS FOR ATOMIC SETS

In this appendix, we give two more extension for the vector case. The first extension is to consider $|\mathcal{A}| = N$, with each $a_j \in \mathcal{A}$ being $\tau$-sparse with non-zero $(index, value)$ pairs $(i_k, (a_j)_k)$, i.e., each $a_j \in \mathbb{R}^d$, but has only $\tau$ non-zero elements. Assume that the non-zero elements are accessed with a quantum oracle $V$ which implements the transformation $V |j\rangle |k\rangle |0\rangle |0\rangle \to |j\rangle |k\rangle |i_k\rangle |(a_j)_k\rangle$. One can construct a coherent access to the non-zero elements

$$V^{\otimes\tau} |j\rangle \bigotimes_{k=1}^\tau |k\rangle |0\rangle |0\rangle = |j\rangle \bigotimes_{k=1}^\tau |k\rangle |i_k\rangle |(a_j)_k\rangle \tag{12}$$

using $\tau$ calls of $V$. Then, a slight modification of the method of Section 3.1 can compute the FW update using $O(\tau\sqrt{N}\log(1/\delta))$ queries to $V$ and $U_g$.

The second extension is to consider latent group norm constraints, which have found use in inducing sparsity in problems in machine learning Jenatton et al. (2011). The $\ell_1$ norm, $d$-simplex, group lasso etc. are all special cases of this.

Following Jaggi (2013) we let $\mathcal{G} = \{\mathfrak{g}_1, \mathfrak{g}_2, \ldots, \mathfrak{g}_{|\mathcal{G}|}\}$, $\mathfrak{g}_i \subseteq [d]$, $\bigcup_i \mathfrak{g}_i = [d]$. Note that the $\mathfrak{g}_i$ need not be disjoint. For each $\mathfrak{g} \in \mathcal{G}$, let $\|\cdot\|_\mathfrak{g}$ be an arbitrary $\ell_p$ norm, and define the **latent group norm**

$$\|x\|_\mathcal{G} := \min_{v_{(\mathfrak{g})} \in \mathbb{R}^{|\mathfrak{g}|}} \sum_{\mathfrak{g} \in \mathcal{G}} \|v_{(\mathfrak{g})}\|_\mathfrak{g}$$

$$\text{s.t.} \quad x = \sum_{\mathfrak{g} \in \mathcal{G}} v_{[\mathfrak{g}]} \tag{13}$$

where $v_{(\mathfrak{g})} \in \mathbb{R}^\mathfrak{g}$ is the restriction of $v \in \mathbb{R}^d$ to coordinates in $\mathfrak{g}$, and $v_{[\mathfrak{g}]} \in \mathbb{R}^d$ has zeros outside the support of $\mathfrak{g}$. In this case, the Frank-Wolfe update corresponds to finding the value $s : \|s\|_\mathcal{G} \leq 1$ such that $s^\top \nabla f(x) = \|\nabla f(x)\|_\mathcal{G}^*$, where $\|\nabla f(x)\|_\mathcal{G}^* = \max_{s:\|s\|_\mathcal{G} \leq 1} s^\top \nabla f(\boldsymbol{x})$.

By Section 4.1 in Jaggi (2013), this norm is an atomic norm, and the dual norm is given by

$$\|\nabla f(x)\|_\mathcal{G}^* = \max_{\mathfrak{g} \in \mathcal{G}} \|\nabla f(x)_{(\mathfrak{g})}\|_\mathfrak{g}^*, \tag{14}$$

which implies that

$$\max_{s:\|s\|_\mathcal{G} \leq 1} (-s^\top \nabla f(x)) = \max_{\mathfrak{g} \in \mathcal{G}} \max_{s:\|s\|_\mathfrak{g} \leq 1} (-s^\top \nabla f(x)). \tag{15}$$

Therefore, it suffices to consider each $\|\cdot\|_\mathfrak{g}$ ball separately, and then do quantum maximizing over all the $|\mathcal{G}|$ balls to find the one that has the largest value of $\|-\nabla f(x)_{\mathfrak{g}_i}\|_{p_i}^*$. The quantum Frank-Wolfe algorithm over latent group norm ball is then given in Algorithm 6. Note that by the absolute homogeneity property of dual norms,

$$\|\nabla f(x)_{(\mathfrak{g})}\|_\mathfrak{g}^* = \|-\nabla f(x)_{(\mathfrak{g})}\|_\mathfrak{g}^*, \tag{16}$$

certain negative signs have been omitted in the algorithmic formulation.

---

**Algorithm 6** Quantum Frank-Wolfe Algorithm over Latent Group Norm Ball

1: **Input:** Gap $\varepsilon$, accuracy $\{\sigma_t\}_{t=1}^T$, iterations $T$.
2: **Initialize:** Let $\boldsymbol{x}^{(1)} \in \mathcal{D}$.
3: **for** $t = 1, ..., T$ **do**
4:   Let $\gamma_t = \frac{2}{t+2}$, $\boldsymbol{x} = \boldsymbol{x}^{(t)}$.
5:   Prepare state $\sum_{i=1}^n |i\rangle_A |\boldsymbol{x}\rangle \bigotimes_{j=1}^{|\mathfrak{g}_i|} |\mathfrak{g}_{i,j}\rangle |0\rangle |0\rangle |0\rangle |0\rangle$.
6:   Perform quantum gradient circuit to get $\sum_{i=1}^n |i\rangle_A |\boldsymbol{x}\rangle \bigotimes_{j=1}^{|\mathfrak{g}_i|} |\mathfrak{g}_{i,j}\rangle |g_{\mathfrak{g}_{i,j}}(\boldsymbol{x})\rangle |0\rangle |0\rangle |0\rangle$,
     where $g_{\mathfrak{g}_{i,j}}(\boldsymbol{x}) = \frac{f(\boldsymbol{x} + \sigma_t \boldsymbol{e}_{\mathfrak{g}_{i,j}}) - f(\boldsymbol{x})}{\sigma_t}$
7:   Compute $\sum_{i=1}^n |i\rangle_A |\boldsymbol{x}\rangle \left( \bigotimes_{j=1}^{|\mathfrak{g}_i|} |\mathfrak{g}_{i,j}\rangle |g_{\mathfrak{g}_{i,j}}(\boldsymbol{x})\rangle \left|\mathsf{sgn}(g_{\mathfrak{g}_{i,j}}(\boldsymbol{x}))|g_{\mathfrak{g}_{i,j}}(\boldsymbol{x})|^{q_i - 1}\right\rangle \right)$
     $\left| \|g(\boldsymbol{x})_{(\mathfrak{g}_i)}\|_{p_i} \right\rangle \left| \|g(\boldsymbol{x})_{(\mathfrak{g}_i)}\|_{p_i}^* \right\rangle$.
8:   Apply quantum maximum finding on the last register, and then measure the rest registers, denote the result of the first register as $i_t$.
9:   Initial $\boldsymbol{s} = 0$, set $\boldsymbol{s}_{\mathfrak{g}_{i_t, j}} = \mathsf{sgn}(g_{\mathfrak{g}_{i_t, j}}(\boldsymbol{x}))|g_{\mathfrak{g}_{i_t, j}}(\boldsymbol{x})|^{q_{i_t} - 1}$ for $j = 1$ to $|\mathfrak{g}_i|$, where $\frac{1}{p_{i_t}} + \frac{1}{q_{i_t}} = 1$. Then normalize $\boldsymbol{s}$.
10:   Update $\boldsymbol{x}^{(t+1)} = (1 - \gamma_t)\boldsymbol{x}^{(t)} + \gamma_t \boldsymbol{s}$.
11: **end for**

---

To simplify the proof of the query complexity of the quantum FW update (Lemma 11), we first assume that the gradient estimation and the maximum-finding are exact, with proof given in Appendix B.6. Then we give the error analysis and show how to choose the parameters $\sigma_t$ in Theorem 6, with proof given in Appendix B.7.

**Lemma 11.** *[Quantum FW update over latent group norm ball] Let $\|\cdot\|_{\mathcal{G}}$ be a latent group norm corresponding to $\mathcal{G} = \{\mathfrak{g}_1, \mathfrak{g}_2, \ldots, \mathfrak{g}_{|\mathcal{G}|}\}$, and let $|\mathfrak{g}|_{\max} = \max_j |\mathfrak{g}_j|$. Then, there exists a quantum algorithm computing the Frank-Wolfe update $s^* := \mathrm{argmax}_{\hat{s} \in \|\cdot\|_{\mathcal{G}}\text{-Ball}} \langle \hat{s}^\top g(\boldsymbol{x}) \rangle$ in $O(\sqrt{|\mathcal{G}|}|\mathfrak{g}|_{\max})$ calls to $\boldsymbol{U}_f$.*

**Theorem 6.** *[Quantum FW over latent group norm ball] By setting $\sigma_t = \frac{C_f}{\sqrt{dL(t+2)}\max_{i\in[|\mathcal{G}|]}|\mathfrak{g}_i|^{1/p_i}}$ for $t \in [T]$, the quantum algorithm (Algorithm 6) solves the latent group norm constraint optimization problem for any precision $\varepsilon$ such that $f(\boldsymbol{x}^T) - f(\boldsymbol{x}^*) \leq \varepsilon$ in $T = \frac{4C_f}{\varepsilon} - 2$ rounds, succeed with probability $1 - p$, with $O\left(\sqrt{|\mathcal{G}|}|\mathfrak{g}|_{\max}\log\frac{C_f}{p\varepsilon}\right)$ calls to the function value oracle $\boldsymbol{U}_f$ per round.*

## A.3 NOTATIONS AND ASSUMPTIONS FOR QUANTUM COMPUTATION

**Basic Notions in Quantum Computing.** Quantum computing utilizes Dirac notation as its mathematical foundation. Let $\{|i\rangle\}_{i=0}^{d-1}$ denote the computational basis of $\mathbb{C}^d$ as $\{|i\rangle\}_{i=0}^{d-1}$, where $|i\rangle$ is a $d$-dimensional unit vector with 1 at the $i^{th}$ position and 0 elsewhere. A $d$-dimensional quantum state is represented as a unit vector $|v\rangle = (v_1, v_2, \ldots, v_d)^T = \sum_i v_i |i\rangle \in \mathbb{C}^d$ with complex amplitudes $v_i$ satisfying $\sum_i |v_i|^2 = 1$.

**Composite Systems.** The joint state of two quantum systems $|v\rangle \in \mathbb{C}^{d_1}$ and $|u\rangle \in \mathbb{C}^{d_2}$ is described by the tensor product $|v\rangle \otimes |u\rangle = (v_1 u_1, v_1 u_2, \ldots v_2 u_1, \ldots, v_{d_1} u_{d_2}) \in \mathbb{C}^{d_1 \times d_2}$ The $\otimes$ symbol is omitted when context permits.

**Quantum Dynamics.** Closed system evolution is described by unitary transformations. Quantum measurement in the computational basis probabilistically projects the state onto a basis vector $|i\rangle$ with the probability of the square of the magnitude of its amplitude. For example, measuring $|v\rangle = \sum_i v_i |i\rangle$ yields outcome $i$ with probability $|v_i|^2$, followed by post-measurement state $|i\rangle$.

**Quantum Access Models.** In general, In quantum computing, access to the objective function is facilitated through quantum oracles $Q_f$, which is a unitary transformation that maps a quantum state $|x\rangle |q\rangle$ to the state $|x\rangle |q + f(x)\rangle$, where $|x\rangle$, $|q\rangle$ and $|q + f(x)\rangle$ are basis states corresponding to the floating-point representations of $x$, $q$ and $q + f(x)$. Moreover, given the superposition input $\sum_{x,q} \alpha_{x,q} |x\rangle |q\rangle$, by linearity the quantum oracle will output the state $\sum_{x,q} \alpha_{x,q} |x\rangle |q + f(x)\rangle$.

## A.4 EXTENDED RELATED WORKS

The Frank-Wolfe (FW) algorithm, also known as the conditional gradient method, has evolved through several key theoretical and applied research phases. The original FW framework Frank et al. (1956) established a projection-free method for quadratic programming with optimal convergence rates when solutions lie on the feasible set boundary, a property later rigorously proven by Canon & Cullum (1968). Wolfe's away-step modification Wolfe (1970) addressed boundary solution limitations, while Dunn's extension Dunn & Harshbarger (1978) generalized FW to smooth optimization over Banach spaces using linear minimization oracles.

Modern convergence analyzes were unified by Jaggi (2013), who introduced duality gap certificates for primal-dual convergence in constrained convex optimization. For strongly convex objectives, Garber & Hazan (2016) demonstrated accelerated linear convergence rates. Projection-free optimization on non-smooth objective functions was studied in Lan (2013); Argyriou et al. (2014); Pierucci et al. (2014). Data-dependent convergence bounds on spectahedrons were improved by Garber (2016) and Allen-Zhu et al. (2017).

Note that the framework was extended to online and stochastic optimizations, inspiring a series of seminal contributions Hazan & Kale (2012); Garber & Hazan (2016); Levy & Krause (2019); Lan & Zhou (2016); Hazan & Luo (2016); Chen et al. (2018); Hassani et al. (2020); Xie et al. (2020); Yurtsever et al. (2019); Zhang et al. (2020). Our future research will explore quantum-enhanced acceleration for these online/stochastic settings. Meanwhile, in recent years, FW methods have gained attention for their effectiveness in dealing with structured constraint problem arising in machine learning and data science, such as LASSO, SVM training, matrix completion and clustering detection. Readers are referred to Bomze et al. (2021); Pokutta (2023) for more information.

The algorithms we develop in the matrix domain belong to the quantum algorithmic family for linear systems. This family originated with the seminal HHL algorithm Harrow et al. (2009), which solves quantum linear systems and achieves exponential speedups over classical methods for well-conditioned sparse matrices. Subsequent improvements reduced dependency on condition number and sparsity Ambainis (2012); Childs et al. (2017); Wossnig et al. (2018). The HHL framework has been successfully adapted to machine learning tasks including support vector machines Rebentrost et al. (2014), supervised and unsupervised machine learning Lloyd et al. (2013), principal component analysis Lloyd et al. (2014) and recommendation systems Kerenidis & Prakash (2017). One can reduce the condition number by preprocessing the matrix itself, and QRAM can help to accelerate such preprocessing. Based on this, the quantum singular value estimation method was developed in Kerenidis & Prakash (2017) and was generalized in Kerenidis & Prakash (2020b). Furthermore, recent work integrates QSVE with state-vector tomography, amplitude amplification/estimation, and spectral norm analysis to enable top-$k$ singular vector extraction Bellante et al. (2022).

Recently, quantum computing has emerged as a promising new paradigm to accelerate a large number of important optimization problems, e.g., combinatorial optimization Grover (1996); Ambainis & Špalek (2006); Dürr et al. (2006); Durr & Hoyer (1996); Mizel (2009); Yoder et al. (2014); Sadowski (2015); He et al. (2020), linear programming Kerenidis & Prakash (2020a); Li et al. (2019); van Apeldoorn & Gilyén (2019b); Apers & Gribling (2023), second-order cone programming Kerenidis et al. (2019c;b;a), quadratic programming Kerenidis & Prakash (2020b), polynomial optimization Rebentrost et al. (2019), semi-definite optimization Kerenidis & Prakash (2020a); van Apeldoorn & Gilyén (2019a); Brandão & Svore (2017); Brandão et al. (2019); van Apeldoorn et al. (2017), convex optimization van Apeldoorn et al. (2020); Chakrabarti et al. (2020); Zhang et al. (2024), nonconvex optimization Zhang & Li (2023); Chen et al. (2025b), stochastic optimization Sidford & Zhang (2023) online optimization He et al. (2022; 2024); Lim & Rebentrost (2022), multi-arm bandit Casalé et al. (2020); Wang et al. (2021); Li & Zhang (2022); Wan et al. (2023). The quantum community is actively pursuing further accelerations of quantum computing in the field of optimization.

### A.5 DISCUSSION OF THE TWO QUANTUM FRANK-WOLFE ALGORITHMS FOR THE MATRIX CASE

We essentially developed two complementary algorithms tailored to high-rank and low-rank gradient matrices, respectively. For Algorithm 3, quantum advantage exists when $d > r/\sqrt{\sigma_1 - \sigma_2}\epsilon$. For Algorithm 4, quantum advantage holds when $d > \sqrt{r}\sqrt{\sigma_1 - \sigma_2}/\epsilon^2(1 - \sigma_1)$. Since the quantum subroutines in the matrix section effectively process the gradient matrix normalized by its Frobenius norm, when this matrix has very low rank, $1 - \sigma_1$ tends to be small (approaching 0 when the rank is 1). In such cases, Algorithm 3 delivers better performance, whereas Algorithm 4 is more suitable otherwise. These two complementary algorithms deliver a quantum speedup of at least $O(\sqrt{d})$.

Furthermore, the repetition steps required for quantum state tomography can be parallelized in the quantum computing cluster. By utilizing $O(d)$ quantum computers simultaneously, the dependence of $d$ in time complexity can be eliminated, giving a parallel time complexity of $\tilde{O}\left(\frac{r\sigma_1^3(M)}{(\sigma_1(M) - \sigma_2(M))\varepsilon^2}\right)$ and $\tilde{O}\left(\frac{\sqrt{r}\sigma_1^4(M)}{(1 - \sigma_1(M))\gamma'^3_{\min}\varepsilon^3}\right)$.

**Remark 1.** *Note that in Section 4, for simplicity of presentation, we focus on square matrices. However, all of the quantum techniques mentioned above can also be applied to non-square matrices, since the quantum singular value estimation can be applied to non-square matricesKerenidis & Prakash (2020b).*

**Remark 2.** *All parameters can be determined during preprocessing. Since tomography constitutes the dominant part of the computational overhead, this preprocessing will not affect the final asymptotic complexity. The choice of $\delta_t$ relates to the maximum singular value of the current gradient matrix. Its range can be determined by running Quantum Singular Value Estimation (QSVE) followed by a maximum-value search algorithm. The purpose of $\epsilon_t$ is to ensure that the ordering of the largest and second-largest singular values does not become misordered during QSVE execution. This parameter can be determined via two methods: 1. During preprocessing, run QSVE-quantum maximum search and perform a binary search to find the critical point where two measurement outcomes appear. Then perform another binary search on $\epsilon_t$ to locate the critical point that distinguishes between these two outcomes. 2. Use the results of amplitude estimation as an indicator to*

*identify the critical point where a sudden jump in amplitude occurs. Since tomography remains the primary source of algorithmic overhead, the computational cost of this process will not impact the final asymptotic complexity.*

**Remark 3.** *Note that both the classical and quantum algorithms in this section assume that the gradients are pre-stored at the memory. In some applications, obtaining the gradients may not be easy, and even directly loading them into the memory would scale linearly with the size of the matrix. This work focuses only on the computation of the update direction, but the gradient calculation time, which is also ignored in classical algorithms Jaggi (2013), is explicitly included in the result Table 2. This is because in quantum computing, there exist several well-established algorithms for gradient estimation Jordan (2005); Gilyén et al. (2019). Moreover, in some applications (such as the matrix completion problem, which we will clarify below), the gradient matrix is sparse. In such applications, the construction of the corresponding quantum memory depends on the sparsity rather than the dimension. The potential acceleration in the gradient calculation and state preparation are left for future exploration.*

To show that solving Equation (2) is a special case of solving Equation (10), let $Z = X/k$. Then, the constraint $\|X\|_{\mathrm{tr}} \leq k$ becomes $\|Z\|_{\mathrm{tr}} = \|X/k\|_{\mathrm{tr}} = \|X\|_{\mathrm{tr}}/k \leq 1$. Substituting into the objective function of Equation (2):

$$\sum_{(i,j)\in\Omega} (X_{i,j} - Y_{i,j})^2 = \sum_{(i,j)\in\Omega} (kZ_{i,j} - Y_{i,j})^2. \tag{17}$$

Define the function $f(Z) = \sum_{(i,j)\in\Omega}(kZ_{i,j} - Y_{i,j})^2$. Then, Equation (2) is equivalent to:

$$\min_{\|Z\|_{\mathrm{tr}}\leq 1} f(Z). \tag{18}$$

This matches the form of Equation (10).

**Satisfaction of Assumption 2.** The trace norm $\|\cdot\|_{\mathrm{tr}}$ is a convex function, so the set $\{Z : \|Z\|_{\mathrm{tr}} \leq 1\}$ is convex. In the finite-dimensional space $\mathbb{R}^{d\times d}$, the set $\{Z : \|Z\|_{\mathrm{tr}} \leq 1\}$ is closed (because the trace norm is continuous) and bounded (since $\|Z\|_F \leq \|Z\|_{\mathrm{tr}} \leq 1$), hence it is compact. For any $Z_1, Z_2 \in D$, we have $\|Z_1\|_F \leq 1$ and $\|Z_2\|_F \leq 1$, so:

$$\|Z_1 - Z_2\|_F \leq \|Z_1\|_F + \|Z_2\|_F \leq 2. \tag{19}$$

Thus, the diameter $D \leq 2$. Therefore, Assumption 2 is satisfied.

**Satisfaction of Assumption 1.** The function $f(Z) = \sum_{(i,j)\in\Omega}(kZ_{i,j} - Y_{i,j})^2$ is a sum of squares, hence it is convex. For $(i,j) \in \Omega$, the partial derivative is $2(kZ_{i,j} - Y_{i,j})$; for $(i,j) \notin \Omega$, it is 0. Therefore, the gradient $\nabla f(Z) = 2P_\Omega(kZ - Y)$, where $P_\Omega$ is the projection operator that preserves elements in $\Omega$ and sets others to zero. For any $Z_1, Z_2$,

$$\nabla f(Z_1) - \nabla f(Z_2) = 2P_\Omega(kZ_1 - kZ_2). \tag{20}$$

Since $P_\Omega$ is a linear operator and does not increase the Frobenius norm, we have

$$\|\nabla f(Z_1) - \nabla f(Z_2)\|_F = \|2P_\Omega(Z_1 - Z_2)\|_F \leq 2k\|Z_1 - Z_2\|_F. \tag{21}$$

Thus, $\nabla f$ is Lipschitz continuous with constant $L = 2k$. Therefore, Assumption 1 is satisfied.

In conclusion, we can apply the algorithms from Section 4 to solve the matrix completion problem. Furthermore, since the gradient of the matrix completion problem is sparse (with only $|\Omega|$ non-zero entries and zeros elsewhere), the construction of quantum memory depends solely on $|\Omega|$ rather than the dimension $d$. Moreover, the computation of the update rule can be further accelerated by leveraging quantum multiplication for sparse matrix. This aspect is left for future investigation.

## A.6 POTENTIAL APPLICATIONS

Our proposed quantum Frank-Wolfe algorithms are applicable to a broad class of convex optimization problems with structured constraints. This section elaborates on the applications of our algorithms in three key domains: sparsity constraints in signal processing, zero-sum games in game theory, and semidefinite programming.

**Signal Processing: Sparsity Constraints via $\ell_1$ Norm.** In signal processing, a common problem is recovering sparse signals from noisy observations, typically achieved through $\ell_1$ norm regularization to promote sparsity in the solution. Consider the basis pursuit denoising problem:

$$\min_{\boldsymbol{x} \in \mathbb{R}^d} \frac{1}{2} \|\boldsymbol{A}\boldsymbol{x} - \boldsymbol{b}\|_2^2 \quad \text{subject to} \quad \|\boldsymbol{x}\|_1 \leq \tau, \tag{22}$$

where $\boldsymbol{A} \in \mathbb{R}^{m \times d}$ is the measurement matrix, $\boldsymbol{b} \in \mathbb{R}^m$ is the observation vector, and $\tau > 0$ is the constraint radius. The feasible domain $\mathcal{D} = \{\boldsymbol{x} \in \mathbb{R}^d : \|\boldsymbol{x}\|_1 \leq \tau\}$ is an $\ell_1$-norm ball. As discussed in Section 3, the core of the Frank-Wolfe update step under this constraint involves solving the linear subproblem $\min_{\boldsymbol{s} \in \mathcal{D}} \langle \boldsymbol{s}, \nabla f(\boldsymbol{x}^{(t)}) \rangle$, whose exact solution is given by the coordinate with the largest absolute gradient component (i.e., $\hat{\boldsymbol{s}} = -\tau \cdot \text{sign}(\nabla_i f(\boldsymbol{x}^{(t)})) \cdot \boldsymbol{e}_i$, where $i = \text{argmax}_j |\nabla_j f(\boldsymbol{x}^{(t)})|$). Our quantum Frank-Wolfe algorithm (Theorem 1) can be use to reduced the per-iteration query complexity from the classical $O(d)$ to $O(\sqrt{d})$.

**Game Theory: Zero-Sum Games with Simplex Constraints.** In game theory, Nash equilibria for two-player zero-sum games can be found by solving a linear programming problem over the simplex. Consider a game with payoff matrix $\boldsymbol{A} \in \mathbb{R}^{m \times n}$. The row player's mixed strategy is a vector $\boldsymbol{x} \in \Delta_m$ ($m$-dimensional simplex), and the column player's mixed strategy is a vector $\boldsymbol{y} \in \Delta_n$. The row player aims to minimize the expected loss $\boldsymbol{x}^\top \boldsymbol{A} \boldsymbol{y}$. Finding the Nash equilibrium can be formulated as:

$$\min_{\boldsymbol{x} \in \Delta_m} \max_{\boldsymbol{y} \in \Delta_n} \boldsymbol{x}^\top \boldsymbol{A} \boldsymbol{y}. \tag{23}$$

Through linear programming duality or its variants, this problem can be transformed into an optimization problem over the simplex. The feasible domain is the simplex $\mathcal{D} = \Delta_d$. The solution to the Frank-Wolfe linear subproblem under this constraint corresponds to the unit vector with the largest gradient component (i.e., $\hat{\boldsymbol{s}} = \boldsymbol{e}_i$, where $i = \text{argmin}_j \nabla_j f(\boldsymbol{x}^{(t)})$). Our quantum Frank-Wolfe algorithm (Theorem 2) similarly accelerates this step, achieving quantum speedup with respect to dimension.

**Semidefinite Programming.** Our quantum algorithms for computing top singular vectors have potential applications in semidefinite programming (SDP). Many SDP solvers, particularly those based on first-order methods, require repeatedly solving linear minimization oracles over the spectrahedron. The solution to this subproblem is given by the outer product of the eigenvector corresponding to the smallest eigenvalue of a symmetric matrix $A$ Nesterov (2007); d'Aspremont (2008); Baes & Bürgisser (2009). Computing this vector is equivalent to finding the top eigenvector of the shifted matrix $-A$. This computational bottleneck is structurally analogous to the top singular vector extraction problem addressed by our quantum subroutines in Section 4. Therefore, our quantum top singular vector extraction (QTSVE) and quantum power method (QPM) algorithms can be integrated into SDP solvers to accelerate this subroutine, providing quantum speedup for a wide class of SDP problems.

# B PROOF DETAIL

## B.1 PROOF OF LEMMA 3

**Lemma 3.** *Given access to the quantum function value oracle $\boldsymbol{U}_f$, there exists a quantum circuit to construct a quantum error bounded gradient oracle $\boldsymbol{U}_g : |i\rangle |\boldsymbol{x}\rangle |0\rangle \rightarrow |i\rangle |\boldsymbol{x}\rangle |g_i(\boldsymbol{x})\rangle$, where $g_i(\boldsymbol{x}) = \frac{f(\boldsymbol{x} + \sigma \boldsymbol{e}_i) - f(\boldsymbol{x})}{\sigma}$ is the $i$-th component of the gradient and $\sigma$ is the tunable parameter, with two queries to the quantum function value oracle.*

**Proof.** By choosing appropriate $\sigma$, we now construct a gradient unitary $\boldsymbol{U}_g : |i\rangle |\boldsymbol{x}\rangle |0\rangle \rightarrow |i\rangle |\boldsymbol{x}\rangle |g_i(\boldsymbol{x})\rangle$ as follows:

$$|i\rangle |\boldsymbol{x}\rangle |0\rangle |0\rangle |0\rangle |0\rangle$$

$$\rightarrow |i\rangle |\boldsymbol{x}\rangle |\boldsymbol{x} + \sigma\boldsymbol{e}_i\rangle |0\rangle |0\rangle |0\rangle \tag{24}$$

$$\rightarrow |i\rangle |\boldsymbol{x}\rangle |\boldsymbol{x} + \sigma\boldsymbol{e}_i\rangle |f(\boldsymbol{x} + \sigma\boldsymbol{e}_i)\rangle |f(\boldsymbol{x})\rangle |0\rangle \tag{25}$$

$$\rightarrow |i\rangle |\boldsymbol{x}\rangle |\boldsymbol{x} + \sigma\boldsymbol{e}_i\rangle |f(\boldsymbol{x} + \sigma\boldsymbol{e}_i)\rangle |f(\boldsymbol{x})\rangle \left|\frac{f(\boldsymbol{x} + \sigma\boldsymbol{e}_i) - f(\boldsymbol{x})}{\sigma}\right\rangle \tag{26}$$

$$\rightarrow |i\rangle |\boldsymbol{x}\rangle \left|\frac{f(\boldsymbol{x} + \sigma\boldsymbol{e}_i) - f(\boldsymbol{x})}{\sigma}\right\rangle \tag{27}$$

$$= |i\rangle |\boldsymbol{x}\rangle |g_i(\boldsymbol{x})\rangle , \tag{28}$$

where Equation (24) is by adding $\sigma$ at the $i$-th entry of the third register, Equation (25) is by applying $\boldsymbol{U}_f$ based on the second and the third register, Equation (26) is by applying addition and division based on the fourth and the fifth register, Equation (27) is by uncomputing the third, fourth and fifth register. For the complexity, this $\boldsymbol{U}_g$ takes two queries of $\boldsymbol{U}_f$ and $O(1)$ elementary gates to get the approximate gradient.

∎

## B.2 PROOF OF LEMMA 4

**Lemma 4.** *(Approximate maximum gradient component finding) Given access to the quantum error bounded gradient oracle $\boldsymbol{U}_g : |i\rangle |\boldsymbol{x}\rangle |0\rangle \rightarrow |i\rangle |\boldsymbol{x}\rangle |g_i(\boldsymbol{x})\rangle$ s.t. for each $i \in [d]$, after measuring $|g_i(\boldsymbol{x})\rangle$, the measured outcome $g_i(\boldsymbol{x})$ satisfies $|g_i(\boldsymbol{x}) - \nabla f_i(\boldsymbol{x})| \leq \epsilon$. There exists a quantum circuit $\mathcal{A}_{\max}$ that finds the index $i^*$ that satisfies $\nabla f_{i^*}(\boldsymbol{x}) \geq \max_{j\in[d]} \nabla f_j(\boldsymbol{x}) - 2\epsilon$ or $|\nabla f_{i^*}(\boldsymbol{x})| \geq \max_{j\in[d]} |\nabla f_j(\boldsymbol{x})| - 2\epsilon$, using $O(\sqrt{d}\log(\frac{1}{\delta}))$ applications of $\boldsymbol{U}_g$, $\boldsymbol{U}_g^\dagger$ and $O(\sqrt{d})$ elementary gates, with probability $1 - \delta$. For the non-uniform initial state, let $p$ be the initial measurement probability of the maximum component, then the algorithm finds the maximum with query complexity of $O(\frac{1}{\sqrt{p}}\log(\frac{1}{\delta}))$.*

**Proof.** We restate the quantum minimum finding algorithm here for reader benefits Durr & Hoyer (1996): Choose threshold index $0 \leq j \leq d - 1$ uniformly at random. Repeat the following and return $j$ when the total running time is more than $22.5\sqrt{d} + 1.4\log(d)$:

1. Prepare the state $\sum_i^d |i\rangle |x\rangle |g_i(x)\rangle |0\rangle$.

2. Set the third register to $|1\rangle$ conditioned on the value of the second register smaller than $g_j(x)$

3. Apply the quantum exponential Grover search algorithm for the third register being $1$.

4. Measure the first and the third registers in computation basis, if the measurement result of the third register is smaller than $g_j(x)$, set $j$ to be the measurement result of the first register.

By Theorem 1 of Durr & Hoyer (1996), the algorithm finds the minimum $g_i(x)$ with probability $1/2$, $O(\sqrt{d})$ applications of $\boldsymbol{U}_g$, $\boldsymbol{U}_g^\dagger$ and $O(\sqrt{d})$ elementary gates. The probability can be boost to $1 - \delta$ with $O(\log(1/\delta))$ repeats and taking the minimum of the outputs.

This algorithm can be modified into the quantum maximum absolute value finding algorithm by setting the third register to $|1\rangle$ conditioned on the value of the second register greater than $|g_j(x)|$ in Step 2, and set $j$ to be the measurement result that is greater than $|g_j(x)|$ in Step 4.

However, with the estimated error, the greatest estimated gradient component $g_{max}(\boldsymbol{x})$ may not have the same index of $\nabla f_{max}(\boldsymbol{x})$. As $|g_i(\boldsymbol{x}) - \nabla f_i(\boldsymbol{x})| \leq \epsilon$ for each $i$, in the worst case, there exists $i$ such that $|g_i(\boldsymbol{x})| = |\nabla f_i(\boldsymbol{x})| + \epsilon \geq |g_{i^*}(\boldsymbol{x})| = \max_{j\in[d]} |\nabla f_j(\boldsymbol{x})| - \epsilon$, the maximum finding algorithm will give such $g_i(\boldsymbol{x})$ as outcome, which is greater than $\max_{j\in[d]} |\nabla f_j(\boldsymbol{x})| - 2\epsilon$.

Similarly, As $|g_i(\boldsymbol{x}) - \nabla f_i(\boldsymbol{x})| \leq \epsilon$ for each $i$, in the worst case, there exists $i$ such that $|g_i(\boldsymbol{x})| = |\nabla f_i(\boldsymbol{x})| - \epsilon \leq |g_{i^*}(\boldsymbol{x})| = \min_{j \in [d]} |\nabla f_j(\boldsymbol{x})| + \epsilon$, the minimum finding algorithm will give such $g_i(\boldsymbol{x})$ as outcome, which is less than $\min_{j \in [d]} |\nabla f_j(\boldsymbol{x})| + 2\epsilon$. Similar proof processes can be employed to derive the error bounds for the minimum/maximum search.

Note that in the matrix case of this work, the state prepared to apply quantum maximum finding is not a uniform superposition, but the algorithm in Durr & Hoyer (1996) is only for the uniform superposition input. For the non-uniform input, in the third step, the Grover operator should be replaced with the amplitude amplification operator. We now prove the complexity of the algorithm for the non-uniform initial state. For the analysis of the probability of success, assume that there is no time-out, that is, the algorithm runs long enough to find the minimum. Then we analyze the probability that an element of a given rank becomes the threshold during the algorithm (Lemma 12) and then bound the expected number of iterations (Lemma 13), which extend Lemma 1 and 2 in Durr & Hoyer (1996).

Then, by Lemma 13, the expected running time of finding the maximum is $O\left(\frac{1}{\sqrt{p_1}}\right)$. By Markov's inequality, after running the algorithm for twice the expected time, the probability of success is at least $1/2$. The probability can be boost to $1 - \delta$ with $O(\log(1/\delta))$ repeats and taking the maximum of the outputs. This extends the Dürr-Høyer minimum finding algorithm to the weighted case and provides a complexity analysis tailored to singular value distributions for the matrix case of this work.

∎

**Lemma 12** (Probability of Selecting Threshold of Rank $r$). *Let $p(t, r)$ be the probability that the element of rank $r$ (where rank 1 is the maximum) will ever be chosen when the infinite algorithm is searching among $t$ elements. Then, for $r \leq t$, $p(t, r) = P_r = \frac{p_r}{\sum_{j=1}^{r} p_j}$, and for $r > t$, $p(t, r) = 0$.*

**Proof**. The case $r > t$ is trivial. For $r \leq t$, we proceed by induction on $t$ for fixed $r$.

**Base step:** When $t = r$, the algorithm starts by measuring the initial state, which yields the element of rank $r$ with probability $P_r$. Since the relative amplitudes of the basis states constituting the marked state remain invariant throughout the amplification process, the probability of selecting rank $r$ as the threshold is exactly $P_r$.

**Inductive step:** Assume that for all $k \in [r, t]$, $p(k, r) = P_r$. Now consider $t + 1$ elements. The initial threshold is chosen with probability $p_r$ for rank $r$. If the initial threshold has rank greater than $r$, then the algorithm will update the threshold only if it finds an element with rank between $r$ and the current threshold. By the induction hypothesis, the probability that rank $r$ is eventually selected when starting from a threshold of rank $k$ (where $r < k \leq t + 1$) is $p(k - 1, r) = P_r$. Therefore,

$$p(t + 1, r) = \frac{p_r}{\sum_{j=1}^{t+1} p_j} + \sum_{k=r+1}^{t+1} \frac{p_k}{\sum_{j=1}^{t+1} p_j} \cdot p(k - 1, r)$$

$$= \frac{1}{\sum_{j=1}^{t+1} p_j} \left( p_r + \sum_{k=r+1}^{t+1} p_k \cdot p(k - 1, r) \right). \tag{29}$$

By the inductive hypothesis,

$$p(t + 1, r) = \frac{1}{\sum_{j=1}^{t+1} p_j} \left( p_r + \sum_{k=r+1}^{t+1} p_k \cdot P_r \right). \tag{30}$$

Substitute $P_r$ into the equation, we have

$$p(t + 1, r) = \frac{1}{\sum_{j=1}^{t+1} p_j} \left( p_r + \sum_{k=r+1}^{t+1} p_k \cdot \frac{p_r}{\sum_{j=1}^{r} p_j} \right). \tag{31}$$

Then, after some simple equivalent transformations, we have

$$p(t+1, r) = \frac{p_r}{\sum_{j=1}^{t+1} p_j} \left( 1 + \frac{\sum_{k=r+1}^{t+1} p_k}{\sum_{j=1}^{r} p_j} \right) = \frac{p_r}{\sum_{j=1}^{t+1} p_j} \left( \frac{\sum_{j=1}^{r} p_j + \sum_{k=r+1}^{t+1} p_k}{\sum_{j=1}^{r} p_j} \right)$$

$$= \frac{p_r}{\sum_{j=1}^{t+1} p_j} \frac{\sum_{j=1}^{t+1} p_j}{\sum_{j=1}^{r} p_j} = \frac{p_r}{\sum_{j=1}^{r} p_j} = P_r \tag{32}$$

This completes the induction. Therefore, the lemma follows. ∎

**Lemma 13** (Expected Running Time). *The expected number of iterations of the quantum maximum finding algorithm for non-uniform initial state is $O\left(\frac{1}{\sqrt{p_1}}\right)$.*

**Proof.** Let $E$ be the expected number of iterations to find the maximum (rank 1). By Lemma 12, the probability that the initial threshold has rank $r$ is $P_r = \frac{p_r}{\sum_{j=1}^{r} p_j}$. When the current threshold has rank $r$, the quantum search algorithm finds a better element (with rank less than $r$) in expected $O(1/\sqrt{S_{r-1}})$ iterations, where $S_{r-1} = \sum_{j=1}^{r-1} p_j$.

Since the threshold rank decreases monotonically, each rank $r$ is visited as a threshold at most once, with probability $P_r$. Thus,

$$E = \sum_{r=1}^{N} P_r \cdot O\left( \frac{1}{\sqrt{S_{r-1}}} \right) = O\left( \sum_{r=2}^{N} \frac{p_r}{S_r} \frac{1}{\sqrt{S_{r-1}}} \right), \tag{33}$$

where $S_r = \sum_{j=1}^{r} p_j$, and for $r = 1$, $S_0 = 0$ and the search time is 0. We have

$$\sum_{r=2}^{N} \frac{p_r}{S_r \sqrt{S_{r-1}}} \leq \sum_{r=2}^{N} \int_{S_{r-1}}^{S_r} x^{-3/2}\, dx = \int_{p_1}^{1} x^{-3/2}\, dx = 2\left( \frac{1}{\sqrt{p_1}} - 1 \right) = O\left( \frac{1}{\sqrt{p_1}} \right), \tag{34}$$

where the first inequality holds because

$$\left( 1 - \sqrt{\frac{S_{r-1}}{S_r}} \right)^2 \geq 0 \implies 1 - \frac{S_{r-1}}{S_r} \leq 2\left( 1 - \sqrt{\frac{S_{r-1}}{S_r}} \right)$$

$$\implies \frac{p_r}{S_r \sqrt{S_{r-1}}} \leq 2\left( \frac{1}{\sqrt{S_{r-1}}} - \frac{1}{\sqrt{S_r}} \right) = \int_{S_{r-1}}^{S_r} x^{-3/2}\, dx. \tag{35}$$

Therefore, $E = O(1/\sqrt{p_1})$, which gives the lemma. ∎

### B.3 PROOF OF THEOREM 1

**Theorem 1.** *(Quantum FW over the sparsity constraint) By setting $\sigma_t = \frac{C_f}{\sqrt{d}L(t+2)}$ for $t \in [T]$, the quantum algorithm (Algorithm 2) solves the sparsity constraint optimization problem for any precision $\varepsilon$ such that $f(\boldsymbol{x}^T) - f(\boldsymbol{x}^*) \leq \varepsilon$ in $T = \frac{4C_f}{\varepsilon} - 2$ rounds, succeed with probability $1 - p$, with $O\left( \sqrt{d} \log \frac{C_f}{p\varepsilon} \right)$ calls to the function value oracle $\boldsymbol{U}_f$ per round.*

**Proof.** By Lemma 2 and the inequality between $\ell_2$ norm and $\ell_\infty$ norm, we have

$$|g_i(\boldsymbol{x}) - \nabla f_i(\boldsymbol{x})| \leq \|g(\boldsymbol{x}) - \nabla f(\boldsymbol{x})\|_\infty \leq \|g(\boldsymbol{x}) - \nabla f(\boldsymbol{x})\|_2 \leq \frac{\sqrt{d}L\sigma}{2}. \tag{36}$$

By Lemma 4, after the quantum approximate maximum absolute value finding, we have an estimated maximum gradient component which satisfied

$$|\nabla f_{i^*}(\boldsymbol{x})| \geq \max_{j \in [d]} |\nabla f_j(\boldsymbol{x})| - \sqrt{d}L\sigma \tag{37}$$

Set $\boldsymbol{s} = -\boldsymbol{e}_{i^*}$, we have

$$\langle \boldsymbol{s}, \nabla f(\boldsymbol{x}^{(t)}) \rangle = -\left| \nabla f_{i^*}(\boldsymbol{x}^{(t)}) \right|$$

$$\leq -\max_{j \in [d]} \left| \nabla f_j(\boldsymbol{x}^{(t)}) \right| + \sqrt{d} L \sigma_t$$

$$= -\langle \boldsymbol{e}_{\mathrm{argmax}_{i \in [d]} |\nabla_i f(\boldsymbol{x}^{(t)})|}, \nabla f(\boldsymbol{x}^{(t)}) \rangle + \sqrt{d} L \sigma_t$$

$$= \min_{\hat{\boldsymbol{s}} \in \mathcal{D}} \langle \hat{\boldsymbol{s}}, \nabla f(\boldsymbol{x}^{(t)}) \rangle + \sqrt{d} L \sigma_t. \tag{38}$$

By the update rule and the definition of the curvature, we have

$$f(\boldsymbol{x}^{(t+1)}) = f((1-\gamma_t)\boldsymbol{x}^{(t)} + \gamma_t \boldsymbol{s}) \leq f(\boldsymbol{x}^{(t)}) + \gamma_t \langle \boldsymbol{s} - \boldsymbol{x}^{(t)}, \nabla f(\boldsymbol{x}^{(t)}) \rangle + \frac{\gamma_t^2}{2} C_f \tag{39}$$

Combining Inequality 38 and 39, we have

$$f(\boldsymbol{x}^{(t+1)}) \leq f(\boldsymbol{x}^{(t)}) + \gamma_t (\min_{\hat{\boldsymbol{s}} \in \mathcal{D}} \langle \hat{\boldsymbol{s}}, \nabla f(\boldsymbol{x}) \rangle - \langle \boldsymbol{x}^{(t)}, \nabla f(\boldsymbol{x}^{(t)}) \rangle) + \sqrt{d} \gamma_t L \sigma_t + \frac{\gamma_t^2}{2} C_f. \tag{40}$$

Let $h(\boldsymbol{x}^{(t)}) := f(\boldsymbol{x}^{(t)}) - f(x^*)$, we have

$$h(\boldsymbol{x}^{(t+1)}) \leq h(\boldsymbol{x}^{(t)}) + \gamma_t (\min_{\hat{\boldsymbol{s}} \in \mathcal{D}} \langle \hat{\boldsymbol{s}}, \nabla f(\boldsymbol{x}) \rangle - \langle \boldsymbol{x}^{(t)}, \nabla f(\boldsymbol{x}^{(t)}) \rangle) + \sqrt{d} \gamma_t L \sigma_t + \frac{\gamma_t^2}{2} C_f$$

$$\leq h(\boldsymbol{x}^{(t)}) - \gamma_t h(\boldsymbol{x}^{(t)}) + \sqrt{d} \gamma_t L \sigma_t + \frac{\gamma_t^2}{2} C_f$$

$$= (1 - \gamma_t) h(\boldsymbol{x}^{(t)}) + \sqrt{d} \gamma_t L \sigma_t + \frac{\gamma_t^2}{2} C_f. \tag{41}$$

Set $\gamma_t = \frac{2}{t+2}, \sigma_t = \frac{\gamma_t C_f}{2\sqrt{d} L}$, we have

$$h(\boldsymbol{x}^{(t+1)}) \leq \left(1 - \frac{2}{t+2}\right) h(\boldsymbol{x}^{(t)}) + \left(\frac{2}{t+2}\right)^2 C_f. \tag{42}$$

Using a similar induction as shown in Jaggi (2013) over $t$, we have

$$h(\boldsymbol{x}^{(t)}) \leq \frac{4C_f}{t+2}. \tag{43}$$

We will restate this induction in Lemma 14 for reader benefit.

Thus, set $\gamma_t = \frac{2}{t+2}, \sigma_t = \frac{C_f}{\sqrt{d} L(t+2)}$ for all $t \in [T]$, after $T = \frac{4C_f}{\varepsilon} - 2$ rounds, we have

$$f(\boldsymbol{x}^{(T)}) - f(x^*) \leq \varepsilon, \tag{44}$$

for any $\varepsilon > 0$.

In each round, by Lemma 3, two queries to the quantum function value oracle are needed to construct the quantum gradient oracle. Then by lemma 4, $O(\sqrt{d} \log \frac{1}{\delta})$ queries to the quantum gradient oracle are needed to find the index of the estimated maximum gradient component with successful probability of $1 - \delta$. Since each maximum finding succeeds with probability $1 - \delta$, the probability that all $T$ iterations succeed is at least $1 - T\delta$. By setting $\delta = p/T$, we ensure an overall success probability of at least $1 - p$. Therefore, $O\left(\sqrt{d} \log \frac{C_f}{p\varepsilon}\right)$ queries to the quantum function value oracle are needed in each iteration. Then the theorem follows.

$$\blacksquare$$

We restate the proof of the induction we use in Theorem 1 for reader benefit.

**Lemma 14.** *(Jaggi (2013)) If for any $t \in [N]$,*

$$h(\boldsymbol{x}^{(t+1)}) \leq \left(1 - \frac{2}{t+2}\right) h(\boldsymbol{x}^{(t)}) + \left(\frac{2}{t+2}\right)^2 C_f, \tag{45}$$

*then*

$$h(\boldsymbol{x}^{(t)}) \leq \frac{4C_f}{t+2}. \tag{46}$$

**Proof.** For $t = 0$, we have

$$h(\boldsymbol{x}^{(1)}) \leq \left(1 - \frac{2}{0+2}\right) h(\boldsymbol{x}^{(0)}) + \left(\frac{2}{0+2}\right)^2 C_f = C_f. \tag{47}$$

Assume that $h(\boldsymbol{x}^{(t)}) \leq \frac{4C_f}{t+2}$, we have

$$
\begin{aligned}
h(\boldsymbol{x}^{(t+1)}) &\leq \left(1 - \frac{2}{t+2}\right) h(\boldsymbol{x}^{(t)}) + \left(\frac{2}{t+2}\right)^2 C_f \\
&\leq \left(1 - \frac{2}{t+2}\right) \frac{4C_f}{t+2} + \left(\frac{2}{t+2}\right)^2 C_f \\
&= \left(1 - \frac{1}{t+2}\right) \frac{4C_f}{t+2} + \left(\frac{2}{t+2}\right)^2 C_f \\
&= \frac{t+1}{t+2} \frac{4C_f}{t+2} \leq \frac{t+2}{t+3} \frac{4C_f}{t+2} = \frac{4C_f}{t+3},
\end{aligned}
\tag{48}
$$

which gives the lemma. ∎

### B.4 PROOF OF LEMMA 10

The framework of quantum gradient estimator originates from Jordan quantum gradient estimation method Jordan (2005), but Jordan algorithm did not give any error bound because the analysis of it was given by omitting the high-order terms of Taylor expansion of the function directly. In 2019, the quantum gradient estimation method with error analysis was given in Gilyén et al. (2019), and was applied to the general convex optimization problem van Apeldoorn et al. (2020); Chakrabarti et al. (2020). In those case, however, $O(\log n)$ repetitions were needed to estimate the gradient within an acceptable error. The query complexity was then improved to $O(1)$ in He et al. (2022; 2024). Here we use the version of He et al. (2024) (Algorithm 7).

---

**Algorithm 7** Bounded-error Jordan quantum gradient estimation He et al. (2024)

---

1: **Input:** point $x$, parameters $r, \rho, \epsilon$.
2: **Output:** $g(x)$

3: Prepare the initial state: $d$ $b$-qubit registers $\left|0^{\otimes b}, 0^{\otimes b}, \ldots, 0^{\otimes b}\right\rangle$ where $b = \log_2 \frac{G\rho}{4\pi d^2 \beta r}$.

  Prepare 1 $c$-qubit register $|0^{\otimes c}\rangle$ where $c = \log_2 \frac{16\pi d}{\rho} - 1$. And prepare $|y_0\rangle = \frac{1}{\sqrt{2^d}} \sum_{a \in \{0,1,\ldots,2^d-1\}} e^{\frac{2\pi i a}{2^d}} |a\rangle$.

4: Apply Hadamard transform to the first $d$ registers.
5: Perform the quantum query oracle $Q_F$ to the first $d + 1$ registers, where $F(u) = \frac{2^b}{2Gr} \left[ f\left(x + \frac{r}{2^b}\left(u - \frac{2^b}{2}\mathbb{1}\right)\right) - f(x) \right]$, and the result is stored in the $(d+1)$th register.
6: Perform the addition modulo $2^c$ operation to the last two registers.
7: Apply the inverse evaluating oracle $Q_F^{-1}$ to the first $d + 1$ registers.
8: Perform quantum inverse Fourier transformations to the first $d$ registers separately.
9: Measure the first $d$ registers in computation bases respectively to get $m_1, m_2, \ldots, m_n$.
10: $g(x) = \widetilde{\nabla} f(x) = \frac{2G}{2^b}\left(m_1 - \frac{2^b}{2}, m_2 - \frac{2^b}{2}, \ldots, m_n - \frac{2^b}{2}\right)^{\mathrm{T}}.$

---

**Lemma 10.** *(Lemma 1 He et al. (2024)) If $f$ is $G$-Lipschitz continues and $L$-smooth convex function and can be accessed by a quantum function value oracle, then there exists an quantum algorithm that for any $r > 0$ and $1 \geq \rho > 0$, gives the estimated gradient $g(x)$, which satisfies*

$$\Pr\left[\|g(x) - \nabla f(x)\|_\infty > 8\pi n^2(n/\rho + 1)Lr/\rho\right] < \rho, \tag{11}$$

*using $O(1)$ applications of $\boldsymbol{U}_f$ and $O(d \log d)$ elementary gates. The space complexity is $O\left(d \log \frac{G\rho}{4\pi d^2 Lr}\right)$.*

**Proof.** The primary additional gate overhead originates from the quantum Fourier transformation (QFT). Each QFT requires $O(\log d)$ elementary gates, and for $d$ such operations, the total additional elementary gate overhead is $O(d \log d)$. Consequently, the additional elementary gate overhead is $O(d \log d)$.

The states after Step 3 will be:

$$\frac{1}{\sqrt{2^n}} \sum_{a \in \{0,1,\ldots,2^n-1\}} e^{\frac{2\pi i a}{2^n}} \left|0^{\otimes b}, 0^{\otimes b}, \ldots, 0^{\otimes b}\right\rangle \left|0^{\otimes c}\right\rangle |a\rangle. \tag{49}$$

After Step 4:

$$\frac{1}{\sqrt{2^{bn+c}}} \sum_{u_1,u_2,\ldots,u_n \in \{0,1,\ldots,2^b-1\}} \sum_{a \in \{0,1,\ldots,2^c-1\}} e^{\frac{2\pi i a}{2^n}} |u_1, u_2, \ldots, u_n\rangle \left|0^{\otimes c}\right\rangle |a\rangle. \tag{50}$$

After Step 5:

$$\frac{1}{\sqrt{2^{bn+c}}} \sum_{u_1,u_2,\ldots,u_n \in \{0,1,\ldots,2^b-1\}} \sum_{a \in \{0,1,\ldots,2^c-1\}} e^{\frac{2\pi i a}{2^n}} |u_1, u_2, \ldots, u_n\rangle |F(u)\rangle |a\rangle. \tag{51}$$

After Step 6:

$$\frac{1}{\sqrt{2^{bn+c}}} \sum_{u_1,u_2,\ldots,u_n \in \{0,1,\ldots,2^b-1\}} \sum_{a \in \{0,1,\ldots,2^c-1\}} e^{2\pi i F(u)} e^{\frac{2\pi i a}{2^n}} |u_1, u_2, \ldots, u_n\rangle |F(u)\rangle |a\rangle. \tag{52}$$

After Step 7:

$$\frac{1}{\sqrt{2^{bn+c}}} \sum_{u_1,u_2,\ldots,u_n \in \{0,1,\ldots,2^b-1\}} \sum_{a \in \{0,1,\ldots,2^c-1\}} e^{2\pi i F(u)} e^{\frac{2\pi i a}{2^n}} |u_1, u_2, \ldots, u_n\rangle \left|0^{\otimes c}\right\rangle |a\rangle. \tag{53}$$

In the following, the last two registers will be omitted:

$$\frac{1}{\sqrt{2^{bn}}} \sum_{u_1,u_2,\ldots,u_n \in \{0,1,\ldots,2^b-1\}} e^{2\pi i F(u)} |u_1, u_2, \ldots, u_n\rangle. \tag{54}$$

And then we simply relabel the state by changing $u \to v = u - \frac{2^b}{2}$:

$$\frac{1}{\sqrt{2^{bn}}} \sum_{v_1,v_2,\ldots,v_n \in \{-2^{b-1},-2^{b-1}+1,\ldots,2^{b-1}\}} e^{2\pi i F(v)} |v\rangle. \tag{55}$$

We denote Formula (55) as $|\phi\rangle$. Let $g = \nabla f(x)$, and consider the idealized state

$$|\psi\rangle = \frac{1}{\sqrt{2^{bn}}} \sum_{v_1,v_2,\ldots,v_n \in \{-2^{b-1},-2^{b-1}+1,\ldots,2^{b-1}\}} e^{\frac{2\pi i g \cdot v}{2G}} |v\rangle. \tag{56}$$

After Step 9, from the analysis of phase estimation Brassard et al. (2002):

$$\Pr\left[\left|\frac{N g_i}{2G} - m_i\right| > e\right] < \frac{1}{2(e-1)}, \forall i \in [n]. \tag{57}$$

Let $e = n/\rho + 1$, where $1 \geq \rho > 0$. We have

$$\Pr\left[\left|\frac{N g_i}{2G} - m_i\right| > n/\rho + 1\right] < \frac{\rho}{2n}, \forall i \in [n]. \tag{58}$$

Note that the difference in the probabilities of measurement on $|\phi\rangle$ and $|\psi\rangle$ can be bounded by the trace distance between the two density matrices:

$$\| |\phi\rangle\langle\phi| - |\psi\rangle\langle\psi| \|_1 = 2\sqrt{1 - |\langle\phi|\psi\rangle|^2} \le 2\| |\phi\rangle - |\psi\rangle \|. \tag{59}$$

Since $f$ is $L$-smooth, we have

$$
\begin{aligned}
F(v) &\le \frac{2^b}{2Gr}[f(x + \frac{rv}{N}) - f(x)] + \frac{1}{2^{c+1}} \\
&\le \frac{2^b}{2Gr}[\frac{r}{2^b}g \cdot v + \frac{L(rv)^2}{2^{2b}}] + \frac{1}{2^{c+1}} \\
&\le \frac{g \cdot v}{2G} + \frac{2^b Lrn}{4G} + \frac{1}{2^{c+1}}.
\end{aligned} \tag{60}
$$

Then,

$$
\begin{aligned}
\| |\phi\rangle - |\psi\rangle \|^2 &= \frac{1}{2^{bn}} \sum_v |e^{2\pi i F(v)} - e^{\frac{2\pi i g \cdot v}{2G}}|^2 \\
&\le \frac{1}{2^{bn}} \sum_v |2\pi i F(v) - \frac{2\pi i g \cdot v}{2G}|^2 \\
&\le \frac{1}{2^{bn}} \sum_v 4\pi^2 (\frac{2^b Lrn}{4G} + \frac{1}{2^{c+1}})^2.
\end{aligned} \tag{61}
$$

Set $b = \log_2 \frac{G\rho}{4\pi n^2 Lr}$, $c = \log_2 \frac{4G}{2^b nLr} - 1$. We have

$$\| |\phi\rangle - |\psi\rangle \|^2 \le \frac{\rho^2}{16n^2}, \tag{62}$$

which implies $\| |\phi\rangle\langle\phi| - |\psi\rangle\langle\psi| \|_1 \le \frac{\rho}{2n}$. Therefore, by the union bound,

$$\Pr\left[\left|\frac{2^b g_i}{2G} - m_i\right| > n/\rho + 1\right] < \frac{\rho}{n}, \forall i \in [n]. \tag{63}$$

Furthermore, there is

$$\Pr\left[\left|g_i - \widetilde{\nabla}_i f(x)\right| > \frac{2G(n/\rho + 1)}{2^b}\right] < \frac{\rho}{n}, \forall i \in [n], \tag{64}$$

as $b = \log_2 \frac{G\rho}{4\pi n^2 Lr}$, we have

$$\Pr\left[\left|g_i - \widetilde{\nabla}_i f(x)\right| > 8\pi n^2(n/\rho + 1)Lr/\rho\right] < \frac{\rho}{n}, \forall i \in [n]. \tag{65}$$

By the union bound, we have

$$\Pr\left[\|g - \widetilde{\nabla}f(x)\|_\infty > 8\pi n^2(n/\rho + 1)Lr/\rho\right] < \rho, \tag{66}$$

which gives the lemma. ∎

### B.5 PROOF OF THEOREM 5

**Theorem 5.** *(Quantum FW with bounded-error Jordan algorithm)* *By setting* $r_t = \frac{\rho C_f}{16\pi d^2(d/\rho + 1)L(t+2)}$ *for* $t \in [T]$, *the quantum algorithm (Algorithm 5) solves the sparsity constraint optimization problem for any precision* $\varepsilon$ *such that* $f(\boldsymbol{x}^T) - f(\boldsymbol{x}^*) \le \varepsilon$ *in* $T = \frac{4C_f}{\varepsilon} - 2$ *rounds, with* $O(1)$ *calls to the function value oracle* $\boldsymbol{U}_f$ *per round.*

**Proof.** By Lemma 10, with probability greater than $\rho$, we have

$$|g_i(\boldsymbol{x}) - \nabla f_i(\boldsymbol{x})| \le \|g(\boldsymbol{x}) - \nabla f(\boldsymbol{x})\|_\infty \le 8\pi d^2(d/\rho + 1)Lr/\rho. \tag{67}$$

Then the maximum component's coordinate of the estimated gradient $i^* = \text{argmax}_{i \in [d]} |g_i(\boldsymbol{x}^{(t)})|$ satisfies

$$|\nabla f_{i^*}(\boldsymbol{x})| \geq \max_{j \in [d]} |\nabla f_j(\boldsymbol{x})| - 16\pi d^2 (d/\rho + 1) Lr/\rho \tag{68}$$

Set $\boldsymbol{s} = -\boldsymbol{e}_{i^*}$, we have

$$\langle \boldsymbol{s}, \nabla f(\boldsymbol{x}^{(t)}) \rangle = -\left|\nabla f_{i^*}(\boldsymbol{x}^{(t)})\right|$$

$$\leq -\max_{j \in [d]} \left|\nabla f_j(\boldsymbol{x}^{(t)})\right| + 16\pi d^2 (d/\rho + 1) Lr/\rho$$

$$= -\langle \boldsymbol{e}_{\text{argmax}_{i \in [d]} |\nabla_i f(\boldsymbol{x}^{(t)})|}, \nabla f(\boldsymbol{x}^{(t)}) \rangle + 16\pi d^2 (d/\rho + 1) Lr/\rho$$

$$= \min_{\hat{\boldsymbol{s}} \in \mathcal{D}} \langle \hat{\boldsymbol{s}}, \nabla f(\boldsymbol{x}^{(t)}) \rangle + 16\pi d^2 (d/\rho + 1) Lr/\rho. \tag{69}$$

By the update rule and the definition of the curvature, we have

$$f(\boldsymbol{x}^{(t+1)}) = f((1 - \gamma_t)\boldsymbol{x}^{(t)} + \gamma_t \boldsymbol{s}) \leq f(\boldsymbol{x}^{(t)}) + \gamma_t \langle \boldsymbol{s} - \boldsymbol{x}^{(t)}, \nabla f(\boldsymbol{x}^{(t)}) \rangle + \frac{\gamma_t^2}{2} C_f \tag{70}$$

Combining Inequality 69 and 70, we have

$$f(\boldsymbol{x}^{(t+1)}) \leq f(\boldsymbol{x}^{(t)}) + \gamma_t (\min_{\hat{\boldsymbol{s}} \in \mathcal{D}} \langle \hat{\boldsymbol{s}}, \nabla f(\boldsymbol{x}) \rangle - \langle \boldsymbol{x}^{(t)}, \nabla f(\boldsymbol{x}^{(t)}) \rangle) + 16\pi d^2 (d/\rho + 1) L\gamma_t r/\rho + \frac{\gamma_t^2}{2} C_f. \tag{71}$$

Let $h(\boldsymbol{x}^{(t)}) := f(\boldsymbol{x}^{(t)}) - f(x^*)$, we have

$$h(\boldsymbol{x}^{(t+1)}) \leq h(\boldsymbol{x}^{(t)}) + \gamma_t (\min_{\hat{\boldsymbol{s}} \in \mathcal{D}} \langle \hat{\boldsymbol{s}}, \nabla f(\boldsymbol{x}) \rangle - \langle \boldsymbol{x}^{(t)}, \nabla f(\boldsymbol{x}^{(t)}) \rangle) + 16\pi d^2 (d/\rho + 1) L\gamma_t r/\rho + \frac{\gamma_t^2}{2} C_f$$

$$\leq h(\boldsymbol{x}^{(t)}) - \gamma_t h(\boldsymbol{x}^{(t)}) + 16\pi d^2 (d/\rho + 1) L\gamma_t r/\rho + \frac{\gamma_t^2}{2} C_f$$

$$= (1 - \gamma_t) h(\boldsymbol{x}^{(t)}) + 16\pi d^2 (d/\rho + 1) L\gamma_t r/\rho + \frac{\gamma_t^2}{2} C_f. \tag{72}$$

Set $\gamma_t = \frac{2}{t+2}, r_t = \frac{\rho \gamma_t C_f}{32\pi d^2 (d/\rho + 1) L}$, we have

$$h(\boldsymbol{x}^{(t+1)}) \leq \left(1 - \frac{2}{t+2}\right) h(\boldsymbol{x}^{(t)}) + \left(\frac{2}{t+2}\right)^2 C_f. \tag{73}$$

Using a similar induction as shown in Jaggi (2013) over $t$, we have

$$h(\boldsymbol{x}^{(t)}) \leq \frac{4C_f}{t+2}. \tag{74}$$

Thus, set $\gamma_t = \frac{2}{t+2}, r_t = \frac{\rho C_f}{16\pi d^2 (d/\rho + 1) L(t+2)}$ for all $t \in [T]$, after $T = \frac{4C_f}{\varepsilon} - 2$ rounds, we have

$$f(\boldsymbol{x}^{(T)}) - f(x^*) \leq \varepsilon, \tag{75}$$

for any $\varepsilon > 0$.

In each round, by Lemma 10, $O(1)$ queries to the quantum function value oracle are needed to get the estimated gradient vector. Subsequent steps no longer require queries to the oracle. Therefore, in each round, $O(1)$ queries to the quantum function value oracle are needed. Then the theorem follows.

∎

### B.6 PROOF OF LEMMA 11

**Lemma 11.** *[Quantum FW update over latent group norm ball] Let $\|\cdot\|_{\mathcal{G}}$ be a latent group norm corresponding to $\mathcal{G} = \{\mathfrak{g}_1, \mathfrak{g}_2, \ldots, \mathfrak{g}_{|\mathcal{G}|}\}$, and let $|\mathfrak{g}|_{\max} = \max_j |\mathfrak{g}_j|$. Then, there exists a quantum algorithm computing the Frank-Wolfe update $s^* := \operatorname{argmax}_{\hat{s} \in \|\cdot\|_{\mathcal{G}}\text{-Ball}} \langle \hat{s}^\top g(\boldsymbol{x}) \rangle$ in $O(\sqrt{|\mathcal{G}|}|\mathfrak{g}|_{\max})$ calls to $\boldsymbol{U}_f$.*

**Proof.** Assume that all $\|\cdot\|_{\mathfrak{g}}$ are $\ell_p$-norms, i.e. $\|\cdot\|_{\mathfrak{g}_i} = \|\cdot\|_{p_i}$ for some $(p_i \in [1, \infty])$, and have quantum access to each $\mathfrak{g}_i = \{\mathfrak{g}_{i,1}, \mathfrak{g}_{i,2}, \ldots, \mathfrak{g}_{i,|\mathfrak{g}_i|}\} \subseteq [d]$ that load $\mathfrak{g}_i$ into quantum registers via

$$U_{\mathcal{G}} |i\rangle_A |0\rangle \to |i\rangle_A |\mathfrak{g}_{i,1}\rangle |\mathfrak{g}_{i,2}\rangle \ldots |\mathfrak{g}_{i,|\mathfrak{g}_i|}\rangle \tag{76}$$

where $A$ is a $\log |\mathcal{G}|$ qubit register. For each $|\mathfrak{g}_{i,j}\rangle$ one can compute an approximation $|g_{\mathfrak{g}_{i,j}}(\boldsymbol{x})\rangle$ to the $\mathfrak{g}_{i,j}$-th component of the gradient at $\boldsymbol{x}$ by the method in Sec. 3.1.

Noting that $\max_{\hat{s} \in \|\cdot\|_p\text{-Ball}} \boldsymbol{s}^\top \boldsymbol{y} := \|\boldsymbol{y}\|_p^*$ and that

$$s^* := \operatorname*{argmax}_{\hat{s} \in \|\cdot\|_p\text{-ball}} \boldsymbol{s}^\top \boldsymbol{y} \tag{77}$$

has components

$$s_i^* \propto \operatorname{sgn}(\boldsymbol{y}_i)|\boldsymbol{y}_i|^{q-1} \tag{78}$$

where $\frac{1}{p} + \frac{1}{q} = 1$, one can compute

$$|i\rangle_A \bigotimes_{j=1}^{|\mathfrak{g}_i|} |\mathfrak{g}_{i,j}\rangle |0\rangle |0\rangle |0\rangle |0\rangle$$

$$\to |i\rangle_A \bigotimes_{j=1}^{|\mathfrak{g}_i|} |\mathfrak{g}_{i,j}\rangle |g_{\mathfrak{g}_{i,j}}(\boldsymbol{x})\rangle |0\rangle |0\rangle |0\rangle$$

$$\to |i\rangle_A \bigotimes_{j=1}^{|\mathfrak{g}_i|} |\mathfrak{g}_{i,j}\rangle |g_{\mathfrak{g}_{i,j}}(\boldsymbol{x})\rangle \left| \operatorname{sgn}(g_{\mathfrak{g}_{i,j}}(\boldsymbol{x}))|g_{\mathfrak{g}_{i,j}}(\boldsymbol{x})|^{q_i-1} \right\rangle |0\rangle |0\rangle$$

$$\to |i\rangle_A \left( \bigotimes_{j=1}^{|\mathfrak{g}_i|} |\mathfrak{g}_{i,j}\rangle |g_{\mathfrak{g}_{i,j}}(\boldsymbol{x})\rangle \left| \operatorname{sgn}(g_{\mathfrak{g}_{i,j}}(\boldsymbol{x}))|g_{\mathfrak{g}_{i,j}}(\boldsymbol{x})|^{q_i-1} \right\rangle \right) \left| \|g(\boldsymbol{x})_{(\mathfrak{g}_i)}\|_{p_i} \right\rangle |0\rangle$$

$$\to |i\rangle_A \left( \bigotimes_{j=1}^{|\mathfrak{g}_i|} |\mathfrak{g}_{i,j}\rangle |g_{\mathfrak{g}_{i,j}}(\boldsymbol{x})\rangle \left| \operatorname{sgn}(g_{\mathfrak{g}_{i,j}}(\boldsymbol{x}))|g_{\mathfrak{g}_{i,j}}(\boldsymbol{x})|^{q_i-1} \right\rangle \right) \left| \|g(\boldsymbol{x})_{(\mathfrak{g}_i)}\|_{p_i} \right\rangle \left| \|g(\boldsymbol{x})_{(\mathfrak{g}_i)}\|_{p_i}^* \right\rangle \tag{79}$$

Apply quantum maximum finding to the last register can then be used to find $s^*$ in $O(\sqrt{|\mathcal{G}|})$ iterations. Each $g_{\mathfrak{g}_{i,j}}(\boldsymbol{x})$ requires 2 queries to $U_f$, totally $O(|\mathfrak{g}_i|)$ queries for a fixed $i$. In the above the index $i$ ranges over $i = 1, 2, \ldots, |\mathcal{G}|$. The query complexity is therefore $O(\sqrt{|\mathcal{G}|}|\mathfrak{g}|_{\max})$, compared with the classical $\sum_{\mathfrak{g} \in \mathcal{G}} |\mathfrak{g}|$. Then the lemma follows. ∎

### B.7 PROOF OF THEOREM 6

**Theorem 6.** *[Quantum FW over latent group norm ball] By setting $\sigma_t = \frac{C_f}{\sqrt{d}L(t+2)\max_{i \in [|\mathcal{G}|]} |\mathfrak{g}_i|^{1/p_i}}$ for $t \in [T]$, the quantum algorithm (Algorithm 6) solves the latent group norm constraint optimization problem for any precision $\varepsilon$ such that $f(\boldsymbol{x}^T) - f(\boldsymbol{x}^*) \leq \varepsilon$ in $T = \frac{4C_f}{\varepsilon} - 2$ rounds, succeed with probability $1 - p$, with $O\left(\sqrt{|\mathcal{G}|}|\mathfrak{g}|_{\max} \log \frac{C_f}{p\varepsilon}\right)$ calls to the function value oracle $\boldsymbol{U}_f$ per round.*

**Proof**. Let the true gradient component be $g_{\mathfrak{g}_{i,j}}(\boldsymbol{x})$, and its estimated value be $\tilde{g}_{\mathfrak{g}_{i,j}}(\boldsymbol{x})$ such that $\left|\tilde{g}_{\mathfrak{g}_{i,j}}(\boldsymbol{x}) - g_{\mathfrak{g}_{i,j}}(\boldsymbol{x})\right| \leq \frac{\sqrt{d}L\sigma}{2}$. According to Step 7 of the algorithm, the dual norm computation involves:

$$\left\|g(\boldsymbol{x})_{(\mathfrak{g}_i)}\right\|_{p_i}^* = \max_{\boldsymbol{s}\in\mathbb{R}^{|\mathfrak{g}_i|}} \left\{ \sum_{j=1}^{|\mathfrak{g}_i|} \boldsymbol{s}_j g_{\mathfrak{g}_{i,j}}(\boldsymbol{x}) \,\middle|\, \|\boldsymbol{s}\|_{q_i} \leq 1 \right\}, \tag{80}$$

where $\frac{1}{p_i} + \frac{1}{q_i} = 1$. The estimated dual norm is:

$$\left\|\tilde{g}(\boldsymbol{x})_{(\mathfrak{g}_i)}\right\|_{p_i}^* = \max_{\boldsymbol{s}\in\mathbb{R}^{|\mathfrak{g}_i|}} \left\{ \sum_{j=1}^{|\mathfrak{g}_i|} \boldsymbol{s}_j \tilde{g}_{\mathfrak{g}_{i,j}}(\boldsymbol{x}) \,\middle|\, \|\boldsymbol{s}\|_{q_i} \leq 1 \right\}. \tag{81}$$

The dual norm error can be decomposed as

$$\left| \left\|\tilde{g}(\boldsymbol{x})_{(\mathfrak{g}_i)}\right\|_{p_i}^* - \left\|g(\boldsymbol{x})_{(\mathfrak{g}_i)}\right\|_{p_i}^* \right| \leq \max_{\|\boldsymbol{s}\|_{q_i}\leq 1} \left| \sum_{j=1}^{|\mathfrak{g}_i|} \frac{s_j \sqrt{d}L\sigma}{2} \right|. \tag{82}$$

By Hölder's inequality, for any $\boldsymbol{s}$ satisfying $\|\boldsymbol{s}\|_{q_i} \leq 1$, let $\delta_{(\mathfrak{g}_i)}$ be the vector in $\mathbb{R}^{\mathfrak{g}_i}$ with all the component being $\frac{\sqrt{d}L\sigma}{2}$ we have

$$\left| \sum_{j=1}^{|\mathfrak{g}_i|} \frac{s_j \sqrt{d}L\sigma}{2} \right| \leq \|\boldsymbol{s}\|_{q_i} \cdot \|\delta_{(\mathfrak{g}_i)}\|_{p_i} \leq \|\delta_{(\mathfrak{g}_i)}\|_{p_i}. \tag{83}$$

Since $\|\delta_{(\mathfrak{g}_i)}\|_{p_i} \leq \frac{\sqrt{d}L\sigma|\mathfrak{g}_i|^{1/p_i}}{2}$, it follows that

$$\left| \left\|\tilde{g}(\boldsymbol{x})_{(\mathfrak{g}_i)}\right\|_{p_i}^* - \left\|g(\boldsymbol{x})_{(\mathfrak{g}_i)}\right\|_{p_i}^* \right| \leq \|\delta_{(\mathfrak{g}_i)}\|_{p_i} \leq \frac{\sqrt{d}L\sigma|\mathfrak{g}_i|^{1/p_i}}{2}. \tag{84}$$

Then, by Lemma 4 and 11, after Step 8, we have

$$\left\|\tilde{g}(\boldsymbol{x})_{(\mathfrak{g}_{i_t})}\right\|_{p_{i_t}}^* \geq \max_{i\in|\mathcal{G}|} \left\|g(\boldsymbol{x})_{(\mathfrak{g}_i)}\right\|_{p_i}^* - \sqrt{d}L\sigma \max_{i\in[|\mathcal{G}|]} |\mathfrak{g}_i|^{1/p_i}, \tag{85}$$

succeed with probability at least $1-\delta$, with query complexity of $O\left(\sqrt{|\mathcal{G}|}|\mathfrak{g}|_{\max} \log\frac{1}{\delta}\right)$. Set $\delta = \frac{p}{T}$ to ensure that this procedure succeeds for all $T$ iterations.

The rest parallels the proof of Theorem 1. Set $\sigma_t = \frac{C_f}{\sqrt{d}L(t+2)\max_{i\in[|\mathcal{G}|]}|\mathfrak{g}_i|^{1/p_i}}$ for all $t \in [T]$, after $T = \frac{4C_f}{\varepsilon} - 2$ rounds, we have

$$f(\boldsymbol{x}^{(T)}) - f(x^*) \leq \varepsilon, \tag{86}$$

for any $\varepsilon > 0$. Then the theorem follows. ∎

### B.8 PROOF OF LEMMA 7

**Lemma 7.** *(Quantum top singular vector extraction) Let there be efficient quantum access to a matrix $M \in R^{d\times d}$, with singular value decomposition $M = \sum_i^d \sigma_i \boldsymbol{u}_i \boldsymbol{v}_i^T$. Define $p = \frac{\sigma_1^2(M)}{\sum_{i=1}^d \sigma_i^2}$. There exist quantum algorithms that with time complexity $O\left(\frac{\|M\|_F \, dpoly\log d}{\sqrt{p}\epsilon\delta^2}\right)$, give the estimated top singular value $\overline{\sigma}_1$ of $M$ to precision $\epsilon$ and the corresponding unit estimated singular vectors $\boldsymbol{u}, \boldsymbol{v}$ to precision $\delta$ such that $\|\boldsymbol{u} - \boldsymbol{u}_{top}\| \leq \delta$, $\|\boldsymbol{v} - \boldsymbol{v}_{top}\| \leq \delta$ with probability at least $1 - 1/poly(d)$.*

**Proof**. Initialize the quantum registers to the uniform superposition state by using Hadamard gates, we have

$$H^{\otimes d} |0\rangle |0\rangle |0\rangle \rightarrow \sum_i^d |i\rangle |0\rangle |0\rangle. \tag{87}$$

By Assumption 4, we can perform the mapping

$$\sum_i |i\rangle |0\rangle |0\rangle \rightarrow \frac{1}{\|M\|_F} \sum_i^d \sum_j^d M_{ij} |i\rangle |j\rangle |0\rangle, \tag{88}$$

in time $\tilde{O}(1)$. Note that

$$\frac{1}{\|M\|_F} \sum_i^d \sum_j^d M_{ij} |i\rangle |j\rangle |0\rangle = \frac{1}{\|M\|_F} \sum_i^k \sigma_i |\boldsymbol{u}_i\rangle |\boldsymbol{v}_i\rangle |0\rangle. \tag{89}$$

Then by the quantum singular estimation algorithm (QSVE, Lemma 5), we have

$$\frac{1}{\|M\|_F} \sum_i^d \sum_j^d M_{ij} |i\rangle |j\rangle |0\rangle \rightarrow \frac{1}{\|\nabla\|_F} \sum_i^k \sigma_i |\boldsymbol{u}_i\rangle |\boldsymbol{v}_i\rangle |\overline{\sigma}_i\rangle, \tag{90}$$

with the cost of $O\left(\frac{\|M\|_F \text{poly} \log d}{\epsilon}\right)$. This process of generating such a state is treated as an oracle which will be invoked multiple times in the quantum maximum finding. This requires that the errors in the estimates of the singular values should be consistent across multiple runs. Note that the randomness of QSVE comes from the quantum phase estimation algorithm, and the QSVE algorithm of Lemma 5 uses a consistent version of phase estimation. This consistency in phase estimation guarantees that the error patterns are reproducible, thereby maintaining uniform errors over repeated oracle calls.

Set $\epsilon \leq (\sigma_1 - \sigma_2)/2$ to ensure that even with the error of singular value estimation, the estimated largest singular value is still larger than the estimated second largest singular value, which can ensure that when we use the quantum maximum finding algorithm, if succeed, we will always get the superposition state corresponding to the largest singular value. By Lemma 4, the cost of finding the largest singular value is $O\left(\frac{1}{\sqrt{p}}\right)$. By Lemma 6, $O(\frac{d \log d}{\delta^2})$ repeats are needed to tomography the corresponding singular vectors of the largest singular value.

Therefore, the overall complexity is $O\left(\frac{\|M\|_F d \text{poly} \log d}{\sqrt{p} \epsilon \delta^2}\right)$. ∎

### B.9 PROOF OF THEOREM 3

**Theorem 3.** *(Quantum FW with QTSVE) By setting $\delta_t = \frac{C_f}{2(t+2)\sigma_1(M_t)}$ and $\epsilon_t \leq (\sigma_1(M_t) - \sigma_2(M_t))/2$ for $t \in [T]$, the quantum algorithm (Algorithm 3) solves the nuclear norm constraint optimization problem for any precision $\varepsilon$ such that $f(X^T) - f(X^*) \leq \varepsilon$ in $T = \frac{4C_f}{\varepsilon} - 2$ rounds, with time complexity $\tilde{O}\left(\frac{r\sigma_1^3(M_t)d}{(\sigma_1(M_t)-\sigma_2(M_t))\varepsilon^2}\right)$ for computing the update direction per round, where $r$ is the rank of the gradient matrix.*

**Proof.** By Lemma 7, set $\epsilon_t \leq (\sigma_1(M) - \sigma_2(M))/2$ to ensure that the quantum maximum finding algorithm, if succeed, will always get the superposition state of the largest singular value. As the QSVE algorithm from Lemma 5 use a consistent version of phase estimation, the estimated error of the singular value will keep unchanged. Thus, we can measure the register of singular value in the computational basic, to check whether the quantum maximum finding succeed, to boost up the success probability. By Lemma 7, we obtain the estimated singular vectors $\boldsymbol{u}, \boldsymbol{v}$, which satisfy $\|\boldsymbol{u} - \boldsymbol{u}_{top}\| \leq \delta_t, \|\boldsymbol{v} - \boldsymbol{v}_{top}\| \leq \delta_t$, with time complexity $O\left(\frac{\|M\|_F d \text{poly} \log d}{\sqrt{p} \epsilon \delta^2}\right)$.

Note that in the matrix case, the linear optimization subproblem of the Frank-Wolfe framework

$$\min_{\hat{S} \in \mathcal{D}} \langle \hat{S}, M_t \rangle \quad \text{s.t.} \quad \text{tr}\left\{\hat{S}\right\} \leq 1 \tag{91}$$

is equivalent to the following problem

$$\min_{\boldsymbol{x}, \boldsymbol{y} \in \mathbb{R}^d} \boldsymbol{x}^\top M_t \boldsymbol{y} \quad \text{s.t.} \quad \|\boldsymbol{x}\|, \|\boldsymbol{y}\| \leq 1. \tag{92}$$

Therefore, since the update direction $S = \boldsymbol{u}^\top \boldsymbol{v}$, the solution quality of the linear subproblem can be bounded with the solution quality of the equivalent problem, that is

$$
\begin{aligned}
\langle S, M_t \rangle - \min_{\hat{S} \in \mathcal{D}} \langle \hat{S}, M_t \rangle &= \langle \boldsymbol{u}^\top \boldsymbol{v}, M_t \rangle - \min_{\hat{S} \in \mathcal{D}} \langle \hat{S}, M_t \rangle \\
&= \boldsymbol{u}^\top M_t \boldsymbol{v} - \min_{\boldsymbol{x}, \boldsymbol{y} \in \mathbb{R}^d} \boldsymbol{x}^\top M_t \boldsymbol{y} \\
&= \boldsymbol{u}^\top M_t \boldsymbol{v} - \boldsymbol{u}_{top}^\top M_t \boldsymbol{v}_{top}.
\end{aligned}
\tag{93}
$$

Then by Lemma 7 and 15, we have

$$
\left| \boldsymbol{u}^\top M_t \boldsymbol{v} - \boldsymbol{u}_{top}^\top M_t \boldsymbol{v}_{top} \right| \le 2\sigma_1(M_t)\delta_t.
\tag{94}
$$

By the update rule and the definition of the curvature, for each round $t$, we have

$$
f(X^{(t+1)}) = f((1 - \gamma_t)X^{(t)} + \gamma_t S) \le f(X^{(t)}) + \gamma_t \langle S - X^{(t)}, M_t \rangle + \frac{\gamma_t^2}{2} C_f
\tag{95}
$$

Combining Inequality 93, 94 and 95, we have

$$
f(X^{(t+1)}) \le f(X^{(t)}) + \gamma_t (\min_{\hat{S} \in \mathcal{D}} \langle \hat{S}, M_t \rangle - \langle X^{(t)}, M_t \rangle) + 2\gamma_t \sigma_1(M_t)\delta_t + \frac{\gamma_t^2}{2} C_f.
\tag{96}
$$

Let $h(X^{(t)}) := f(X^{(t)}) - f(X^*)$, we have

$$
\begin{aligned}
h(X^{(t+1)}) &\le h(X^{(t)}) + \gamma_t (\min_{\hat{S} \in \mathcal{D}} \langle \hat{S}, M_t \rangle - \langle X^{(t)}, M_t \rangle) + 2\gamma_t \sigma_1(M_t)\delta_t + \frac{\gamma_t^2}{2} C_f \\
&\le h(X^{(t)}) - \gamma_t h(X^{(t)}) + 2\gamma_t \sigma_1(M_t)\delta_t + \frac{\gamma_t^2}{2} C_f \\
&= (1 - \gamma_t)h(X^{(t)}) + 2\gamma_t \sigma_1(M_t)\delta_t + \frac{\gamma_t^2}{2} C_f.
\end{aligned}
\tag{97}
$$

Set $\gamma_t = \frac{2}{t+2}, \delta_t = \frac{\gamma_t C_f}{4\sigma_1(M_t)}$, we have

$$
h(X^{(t+1)}) \le \left( 1 - \frac{2}{t+2} \right) h(X^{(t)}) + \left( \frac{2}{t+2} \right)^2 C_f.
\tag{98}
$$

Using a similar induction as shown in Jaggi (2013) over $t$, we have

$$
h(x^{(t)}) \le \frac{4C_f}{t+2}.
\tag{99}
$$

In summary, set $\gamma_t = \frac{2}{t+2}, \delta_t = \frac{C_f}{2(t+2)\sigma_1(M_t)}$, after $T = \frac{4C_f}{\varepsilon} - 2$ rounds, we have

$$
f(\boldsymbol{x}^{(T)}) - f(\boldsymbol{x}^*) \le \varepsilon,
\tag{100}
$$

for any $\varepsilon > 0$. Since $\delta_t = \frac{C_f}{2(t+2)\sigma_1(M_t)} \ge \frac{C_f}{2(T+2)\sigma_1(M_t)} = \frac{\varepsilon}{2\sigma_1(M)}$, in each round, the time complexity of update computing is $O\left( \frac{\|M\|_F \sigma_1^2(M) d \cdot \text{poly} \log d}{\sqrt{p}(\sigma_1(M) - \sigma_2(M))\varepsilon^2} \right)$. Since $\|M\|_F \le \sqrt{r}\sigma_1(M), p \ge \frac{1}{r}$, the time complexity is upper bounded by $O\left( \frac{r\sigma_1^3(M) d \cdot \text{poly} \log d}{(\sigma_1(M) - \sigma_2(M))\varepsilon^2} \right)$, where $r$ is the rank of the gradient matrix. ∎

**Lemma 15.** *For any* $\|\boldsymbol{x} - \boldsymbol{x}'\|_2, \|\boldsymbol{y} - \boldsymbol{y}'\|_2 \le \delta < 1, \|\boldsymbol{x}\|, \|\boldsymbol{y}\| \le 1$, *we have*

$$
\left| \boldsymbol{x}^\top M \boldsymbol{y} - \boldsymbol{x'}^\top M \boldsymbol{y}' \right| \le 2\sigma_1(M)\delta.
\tag{101}
$$

**Proof.** Since $\boldsymbol{x}^\top M \boldsymbol{y} - \boldsymbol{x'}^\top M \boldsymbol{y}' = (x - x')^\top M \boldsymbol{y} + x'M(\boldsymbol{y} - \boldsymbol{y}')$, we have

$$
\left| \boldsymbol{x}^\top M \boldsymbol{y} - \boldsymbol{x'}^\top M \boldsymbol{y}' \right| \le \sigma_1(M)\|\boldsymbol{x} - \boldsymbol{x}'\|_2 \|\boldsymbol{y}\|_2 + \sigma_1(M)\|\boldsymbol{x}'\|_2 \|\boldsymbol{y} - \boldsymbol{y}'\|_2.
\tag{102}
$$

Thus, for any $\|\boldsymbol{x} - \boldsymbol{x}'\|_2, \|\boldsymbol{y} - \boldsymbol{y}'\|_2 \le \delta < 1, \|\boldsymbol{x}\|, \|\boldsymbol{y}\| \le 1$, we have

$$
\left| \boldsymbol{x}^\top M \boldsymbol{y} - \boldsymbol{x'}^\top M \boldsymbol{y}' \right| \le 2\sigma_1(M)\delta.
\tag{103}
$$

∎

## B.10 PROOF OF LEMMA 9

**Lemma 9.** *(Quantum power method) Let there be quantum access to the matrix $M \in R^{d \times d}$ with $\sigma_{max} \leq 1$, and to a vector $\boldsymbol{z} \in R^d$. Let $\gamma'_{\min}$ be the lower bound of $\left\| (M^\top M)^i \boldsymbol{z} \right\|$ for all $i \in [k]$. There exists a quantum algorithm that creates a state $|\boldsymbol{y}\rangle$ such that $\left\| |\boldsymbol{y}\rangle - |(M^\top M)^k \boldsymbol{z}\rangle \right\| \leq \delta$ in time $\tilde{O}(\frac{k}{\gamma'_{\min}} \|M\|_F \log(1/\delta))$, with probability at least $1 - O(k/poly(d))$.*

**Proof.** Suppose $\|z_l - M z_{l-1}\| \leq \epsilon$ with $z_l = M z_{l-1}$ for $l \in [L]$, and $z_0 = x$, we have

$$\|z_1 - Mx\| \leq \epsilon$$
$$\|z_2 - M^2 x\| \leq \|z_2 - M z_1 + M z_1 - M^2 x\|$$
$$\leq \|z_2 - M z_1\| + \|M z_1 - M^2 x\|$$
$$\leq \epsilon + \|M(z_1 - Mx)\|$$
$$\leq \epsilon + \sigma_{\max}\|z_1 - Mx\|$$
$$\leq (1 + \sigma_{\max})\epsilon$$
$$\|z_3 - M^3 x\| \leq \|z_3 - M z_2 + M z_2 - M^3 x\|$$
$$\leq \|z_3 - M z_2\| + \|M z_2 - M^3 x\|$$
$$\leq \epsilon + \|M(z_2 - M^2 x)\|$$
$$\leq \epsilon + \sigma_{\max}\|z_2 - M^2 x\|$$
$$\leq \epsilon + \sigma_{\max}(1 + \sigma_{\max})\epsilon$$
$$\leq (1 + \sigma_{\max} + \sigma_{\max}^2)\epsilon. \tag{104}$$

We use $\sigma_{\min}\|x\| \leq \|Mx\| \leq \sigma_{\max}\|x\|$, where $\sigma_{\max} = \max_{x \neq 0} x^\top M x / \|x\|^2$. By induction, we have

$$\left\| z_L - M^L x \right\| \leq \sum_{i \in [L]} \sigma_{\max}^{i-1} \epsilon = \frac{\sigma_{\max}^L - 1}{\sigma_{\max} - 1} \epsilon. \tag{105}$$

Let $\gamma'_{\min}$ be the lower bound of $\left\| (M^\top M)^i \boldsymbol{z} \right\|$ for all $i \in [k]$. As each multiplication requires time complexity of $\tilde{O}(\frac{1}{\gamma}\|M\|_F \log(1/\epsilon))$ (Lemma 8), $k$ steps of multiplication require time complexity of $\tilde{O}\left(\frac{k}{\gamma'_{\min}}\|M\|_F \log(1/\epsilon)\right)$. Furthermore, since

$$\log \frac{1 - \sigma_1^k(M)}{1 - \sigma_1(M)} \leq -\log(1 - \sigma_1(M)) \leq \frac{1}{1 - \sigma_1(M)}, \tag{106}$$

if we want $\left\| z_k - M^k x \right\| \leq \delta$, the time complexity will be $\tilde{O}\left(\frac{k\|M\|_F}{(1 - \sigma_1(M))\gamma'_{\min}} \log(1/\delta)\right)$.

∎

## B.11 PROOF OF THEOREM 4

**Theorem 4.** *(Quantum FW with QPM) By setting $k_t = \frac{2C_0 \sigma_1(M_t) \ln d}{\varepsilon}, \delta_t = \delta'_t = \frac{\varepsilon \gamma'_{\min}}{16 \sigma_1(M_t)}$ for $t \in [T]$, the quantum algorithm (Algorithm 4) solves the nuclear norm constraint optimization problem for any precision $\varepsilon$ such that $f(X^T) - f(X^*) \leq \varepsilon$ in $T = \frac{4C_f}{\varepsilon} - 2$ rounds, with time complexity $\tilde{O}\left(\frac{\sqrt{r}\sigma_1^4(M_t)d}{(1 - \sigma_1(M_t))\gamma'^3_{\min}\varepsilon^3}\right)$ for computing the update direction per round, where $r$ is the rank of the gradient matrix, $C_0$ is a constant and $\gamma'_{\min}$ is the lower bound of $\left\| (M_t^\top M_t)^i \boldsymbol{b}) \right\|$ for all $i \in [k]$.*

**Proof.** Denote $(MM^\top)^k \boldsymbol{b}$ as $\boldsymbol{z}_u$, $(M^\top M)^k \boldsymbol{b}$ as $\boldsymbol{z}_v$. For the quantum power method, we first use the Lemma 8 to construct a unitary $U_1$ which computes $k$ steps of multiplication: $U_1 : |\boldsymbol{b}\rangle |\boldsymbol{b}\rangle \to |\bar{\boldsymbol{z}}_u\rangle |\bar{\boldsymbol{z}}_v\rangle$ with $\|\bar{\boldsymbol{z}}_u - \boldsymbol{z}_u\|_2 \leq \delta$ and $\|\bar{\boldsymbol{z}}_v - \boldsymbol{z}_v\|_2 \leq \delta$ (Lemma 9). Then we tomography $|\bar{\boldsymbol{z}}_u\rangle |\bar{\boldsymbol{z}}_v\rangle$ to get $\boldsymbol{u}, \boldsymbol{v}$. Simalar to the proof of Theorem 3, our goal is to ensure $\left| \frac{\boldsymbol{u}^\top M \boldsymbol{v}}{\|\boldsymbol{u}\|\|\boldsymbol{v}\|} - \sigma_1(M) \right| \leq \varepsilon$.

First, we settle down $k = \frac{2C_0 \sigma_1(M) \ln d}{\varepsilon}$ so that we have

$$\left| \frac{\boldsymbol{z}_u^\top M \boldsymbol{z}_v}{\|\boldsymbol{z}_u\| \|\boldsymbol{z}_v\|} - \sigma_1(M) \right| \leq \varepsilon/2. \tag{107}$$

Suppose $\sigma_1(M) < 1$ and $\left\| (M^\top M)^i \boldsymbol{b} \right\| \in [\gamma'_{\min}, 1]$ for $i = 1, ..., k$, after applying $k$ times of quantum matrix-vector multiplication ($U_1$) as described by Lemma 8, we obtain $|\overline{\boldsymbol{z}}_u\rangle |\overline{\boldsymbol{z}}_v\rangle$ with $\|\overline{\boldsymbol{z}}_u - \boldsymbol{z}_u\|_2 \leq \delta$ and $\|\overline{\boldsymbol{z}}_v - \boldsymbol{z}_v\|_2 \leq \delta$ in time $T(U_1) = \tilde{O}\left( \frac{k\|M\|_F}{(1-\sigma_1(M))\gamma'_{\min}} \log(1/\delta) \right)$. Using $U_1$, we can tomography $|\overline{\boldsymbol{z}}_u\rangle |\overline{\boldsymbol{z}}_v\rangle$ and obtain $\boldsymbol{u}, \boldsymbol{v}$ with $\|\boldsymbol{u} - \overline{\boldsymbol{z}}_u\| \leq \delta', \|\boldsymbol{v} - \overline{\boldsymbol{z}}_v\| \leq \delta'$ in time $O\left( \frac{T(U_1) d \log d}{(\delta')^2} \right)$. By the triangle inequality, we have

$$\|\boldsymbol{u} - \boldsymbol{z}_u\|_2 \leq \delta + \delta' \leq 1, \|\boldsymbol{v} - \boldsymbol{z}_v\|_2 \leq \delta + \delta' \leq 1 \tag{108}$$

Notice that

$$\left\| \frac{\boldsymbol{u}}{\|\boldsymbol{u}\|} - \frac{\boldsymbol{z}_u}{\|\boldsymbol{z}_u\|} \right\| = \left\| \frac{\boldsymbol{u}}{\|\boldsymbol{u}\|} - \frac{\boldsymbol{u}}{\|\boldsymbol{z}_u\|} + \frac{\boldsymbol{u}}{\|\boldsymbol{z}_u\|} - \frac{\boldsymbol{z}_u}{\|\boldsymbol{z}_u\|} \right\|$$

$$\leq \left\| \frac{\boldsymbol{u}}{\|\boldsymbol{u}\|} - \frac{\boldsymbol{u}}{\|\boldsymbol{z}_u\|} \right\| + \left\| \frac{\boldsymbol{u}}{\|\boldsymbol{z}_u\|} - \frac{\boldsymbol{z}_u}{\|\boldsymbol{z}_u\|} \right\|$$

$$\leq 2 \frac{\|\boldsymbol{u} - \boldsymbol{z}_u\|}{\|\boldsymbol{z}_u\|}, \tag{109}$$

we have

$$\left\| \frac{\boldsymbol{u}}{\|\boldsymbol{u}\|} - \frac{\boldsymbol{z}_u}{\|\boldsymbol{z}_u\|} \right\| \leq 2 \frac{\delta + \delta'}{\gamma'_{\min}}. \tag{110}$$

Similarly, we have

$$\left\| \frac{\boldsymbol{v}}{\|\boldsymbol{v}\|} - \frac{\boldsymbol{z}_v}{\|\boldsymbol{z}_v\|} \right\| \leq 2 \frac{\delta + \delta'}{\gamma'_{\min}}. \tag{111}$$

Thus, we have

$$\left| \frac{\boldsymbol{u}^\top M \boldsymbol{v}}{\|\boldsymbol{u}\| \|\boldsymbol{v}\|} - \frac{\boldsymbol{z}_u^\top M \boldsymbol{z}_v}{\|\boldsymbol{z}_u\| \|\boldsymbol{z}_v\|} \right| \leq \left| \frac{\boldsymbol{u}^\top M \boldsymbol{v}}{\|\boldsymbol{u}\| \|\boldsymbol{v}\|} - \frac{\boldsymbol{u}^\top M \boldsymbol{z}_v}{\|\boldsymbol{u}\| \|\boldsymbol{z}_v\|} \right| + \left| \frac{\boldsymbol{u}^\top M \boldsymbol{z}_v}{\|\boldsymbol{u}\| \|\boldsymbol{z}_v\|} - \frac{\boldsymbol{z}_u^\top M \boldsymbol{z}_v}{\|\boldsymbol{z}_u\| \|\boldsymbol{z}_v\|} \right|$$

$$\leq \frac{\|M\| \|\boldsymbol{v} - \boldsymbol{z}_v\|}{\|\boldsymbol{z}_v\|} + \frac{\|M\| \|\boldsymbol{u} - \boldsymbol{z}_u\|}{\|\boldsymbol{z}_u\|}$$

$$\leq 4 \frac{(\delta + \delta') \sigma_1(M)}{\gamma'_{\min}}. \tag{112}$$

The remaining proof is similar to that of Theorem 3. Now we set $\delta = \delta' = \frac{\varepsilon \gamma'_{\min}}{16 \sigma_1(M)}$, $\left| \frac{\boldsymbol{u}^\top M \boldsymbol{v}}{\|\boldsymbol{u}\| \|\boldsymbol{v}\|} - \frac{\boldsymbol{z}_u^\top M \boldsymbol{z}_v}{\|\boldsymbol{z}_u\| \|\boldsymbol{z}_v\|} \right| \leq \varepsilon/2$. Therefore, $\left| \frac{\boldsymbol{u}^\top M \boldsymbol{v}}{\|\boldsymbol{u}\| \|\boldsymbol{v}\|} - \sigma_1(M) \right| \leq \varepsilon$. The time complexity is $\tilde{O}(T(U_1) d / (\delta')^2) = \tilde{O}(\frac{\sqrt{r} \sigma_1^4(M) d}{(1-\sigma_1(M)) \gamma'^3_{\min} \varepsilon^3})$, where $r$ is the rank of the gradient matrix. $\blacksquare$

## C   ETHICS STATEMENT

This work is a theoretical study. As such, we do not foresee any immediate specific ethical issues arising from this work. We have conducted our research with integrity and in accordance with the academic standards of our community.

## D   REPRODUCIBILITY STATEMENT

As a theoretical paper, all claims and results are supported by detailed mathematical proofs provided in the main text and appendices. Therefore, the results can be reproduced by verifying the logical steps of the proofs. We have endeavored to make our proofs as clear and self-contained as possible to facilitate verification by the reader.

# E    LLM USAGE

Large Language Models (LLMs) were used solely to aid or polish writing. This includes polishing sentences, improving grammar, and enhancing the readability and fluency of the text.

