# OpenReview forum: "Quantum Algorithms for Projection-Free Sparse Convex Optimization"
_ICLR.cc/2026/Conference — Submitted to ICLR 2026_

### Official Review · Reviewer_jxNP · 2025-10-19

**Soundness:** 3
**Presentation:** 3
**Contribution:** 3
**Rating:** 6
**Confidence:** 4

**Summary:**

This paper provides a quantum version of the Frank-Wolfe algorithm for constrained convex optimization. Compared to prior work (Chen & de Wolf, 2023) which deals with regression with l1 and l2 norm constraints, the authors tackles the convex optimization in a more general setting, proposing methods for optimization on a smooth convex function with l1 norm constraints, a simplex constraint or a latent group norm constraint, as well as matrix nuclear norm constraints. Their algorithm can provide quadratic speedups in terms of the dimension of the domain.

**Strengths:**

1. The paper addresses an interesting problem in quantum optimization and generalizes results to a much larger set of convex optimization problems, extending the applicability of such algorithms from regression to other potential fields such as signal processing (sparsity constraints via l1 norm), game theory (zero sum games with simplex) and SDPs (nuclear norm optimization).
2. Proofs and assumptions are clearly documented, with different scenarios of gradient obtainment for the vector case discussed in detail.

**Weaknesses:**

1. Cost of additional qubits and gates not provided for Theorem 6 in Table I.
2. Applications of this paper can be more clearly articulated, i.e. what's been listed in strengths 1.
3. Assumption for gradient obtainment in the matrix case seems a bit strong, as the gradient has to be low rank _and_ accessible in a KP tree.

**Questions:**

1. Given that some previous quantum algorithms that tackle the problem of convex optimization rely on the multiplicative weight update method eg. quantum LP and SDP solvers, I am curious as to how the quantum Frank-Wolfe method proposed in this methods compares to these methods and what potential advantages/disadvantages exist for the quantum Frank-Wolfe method.
2. Why is a different assumption for gradients used for the matrix case different from the vector case? Is the gradient estimation algorithm not applicable to matrix cases?
3. I think the strength of gradient assumption in the matrix case may be stronger than the membership and separation oracles in ariXiv:1809.00643 as you could obtain such oracles when you have both function and gradient access. How would the algorithm perform this weaker set of assumptions is provided instead of the gradient assumption?

---

> ### Author Response · Authors · 2025-11-20
>
> We sincerely thank the reviewer for their positive assessment of our work's generalization to broader convex optimization settings and their affirmation of our clearly documented proofs and assumptions. Below we provide a point-by-point response to the concerns and questions.
>
> **For weakness 1:**
>
> Thank you for pointing this out. We will supplement the quantum resource costs for Theorem 6 in the revised version:
> Qubits: $O(d + \log |\mathcal{G}| + |\mathfrak{g}|_{\max} \log (1/\varepsilon))$
>
> Gates: $\tilde{O}(\sqrt{|\mathcal{G}|} \cdot |\mathfrak{g}|_{\max})$
>
> **For weakness 2:**
>
> We thank the reviewer for this valuable suggestion and for providing these rich application scenarios. In the revised version, we will list these applications in the introduction and add a subsection titled "Potential Applications" in the Appendix to articulate the practical relevance of our quantum Frank-Wolfe framework. We noticing that the BQP problems mentioned by Reviewer CRBK can benefit from our proposed QTSVE and QPM techniques, we will also include this in the applications.
>
> **For weakness 3:**
>
> We thank the reviewer for raising this point. We clarify that the gradient matrix need not be low-rank. Our two matrix algorithms are complementary: Algorithm 3 (QTSVE) is advantageous for low-rank gradients, while Algorithm 4 (QPM) becomes more favorable for high-rank cases, where a larger $1-\sigma_1$ often occurs. This design ensures coverage across different matrix structures.
>
> **For question 1:**
>
> We thank the reviewer for raising this insightful comparison to quantum multiplicative weight update (MWU) methods. While quantum MWU methods provide a powerful general framework for problems like semidefinite programs (SDPs), they face significant practical challenges when applied to the structured convex optimization problems that are the focus of our work. Specifically, for problems with sparsity-inducing constraints:
> 1. QFW operates directly on the natural problem formulation, while MWU requires an indirect and often costly reformulation of our constraint sets into SDP form, increasing the problem dimensionality.
> 2. QFW leverages efficient quantum primitives like maximum finding and top singular vector extraction, which naturally align with the linear minimization oracle for our constraint sets. In contrast, MWU relies on quantum Gibbs sampling, which depends critically on Hamiltonian condition numbers and incurs higher constant factors.
> 3. The QFW approach specifically exploits the problem structure of sparsity-inducing constraints, providing a more direct and potentially more implementable acceleration pathway. MWU, while offering superior theoretical convergence for certain SDPs, cannot leverage these specific structural advantages as effectively.
>
> **For question 2:**
>
> For the matrix case, we could indeed assume the same function value oracle and naturally employ an improved Jordan's algorithm to achieve a query complexity advantage in gradient estimation. However, in the matrix case, we aimed to further investigate whether quantum algorithms can accelerate the computational complexity of the update step beyond just query counts. By adopting a different input model, we are able to leverage algorithms from the quantum linear algebra toolkit to obtain results for matrix variables.
>
> **For question 3:**
>
> We think, in terms of query complexity, a significant advantage could be achieved in this setting, for reasons analogous to those discussed in our response to Question 2. However, a direct application of a quantum framework of general convex optimization in ariXiv:1809.00643 under these weak oracles would likely lead to a gate complexity that scales cubically with the dimension. Therefore, whether sparsity-inducing constraints can be effectively utilized under weak oracles presents a very interesting direction for future work.
>
> We truly appreciate the reviewer's thoughtful engagement with our work. The questions raised have helped us better contextualize our contributions within the broader landscape of quantum optimization. We will incorporate these valuable discussions into the revised version of our paper. We hope that our responses have adequately addressed all points of concern.

---

> > ### Comment · Reviewer_jxNP · 2025-11-27
> >
> > Thank you to the authors for their response. I will be retaining my positive evaluation of the paper.

---

### Official Review · Reviewer_CRBK · 2025-10-19

**Soundness:** 2
**Presentation:** 3
**Contribution:** 2
**Rating:** 4
**Confidence:** 3

**Summary:**

This work presents a framework for Quantum Frank–Wolfe (QFW) algorithms, extending classical projection-free convex optimization to the quantum setting. The authors design quantum analogues of the Frank–Wolfe method for both vector-domain (e.g., $\ell_1$-ball, simplex constraints) and matrix-domain problems (e.g., trace-norm constraints). By leveraging quantum primitives such as Dürr–Høyer maximum finding, Jordan’s quantum gradient estimation, and quantum singular value estimation (QTSVE), they show their algorithms achieve a $\sqrt{d}$ improvement in dimensional dependence while maintaining the classical convergence rate of $O(1/\varepsilon)$.

**Strengths:**

1. The manuscript presents proofs and complexity analysis, which seem reasonable.

2. The proposed quantum framework attempts to address a range of convex optimization problems in both vector and matrix domains, including $\ell_1$ norm and trace norm (nuclear norm) constraints.

3. The manuscript presents a theoretical exploration of quantum algorithms in the context of convex optimization problems.

**Weaknesses:**

1. The lack of simulations and qubit estimates for the core matrix algorithms (QPM/QTSVE) makes it challenging to assess the practical feasibility and actual performance of the proposed quantum speedups.

2. The algorithm requires strong quantum oracle access (e.g., normalized row states and function value oracles), which are currently not feasible on NISQ devices.

3. The algorithmic components (quantum maximum finding, SVD estimation, gradient estimation) are not new.

4. The use of $\tilde{O}$  notation hides constants that could have a significant impact on the practical runtime and performance of the algorithm.

**Questions:**

1. The QFW algorithm is designed for convex optimization, do you think its techniques could be adapted to solve non-convex problems like BQP, or provide useful insights for tackling such NP-hard problems?

2. How does the quantum acceleration of the Frank-Wolfe algorithm for sparse convex optimization  compare with the optimization methods for bi-quadratic programming over unit spheres (as discussed in [1])  in terms of computational efficiency and the ability to handle high-dimensional optimization problems?




[1].  Li, S., et al., Tighter bound estimation for efficient biquadratic optimization over unit spheres, Journal of Global Optimization, 2024.

---

> ### Author Response · Authors · 2025-11-20
>
> We sincerely thank the reviewer for their thoughtful and constructive feedback on our manuscript. We are particularly grateful for their recognition of our work's strengths, including the theoretical proofs, complexity analysis, and the extension of the quantum Frank-Wolfe framework to both vector and matrix domains. The insightful questions raised have provided us with an excellent opportunity to further clarify the scope and potential impact of our work.
>
> **For weakness 1:**
>
> The quantum circuits of QSVE and QPM require only $O(\log d)$ qubits. Furthermore, if we have $O(d \log d)$ qubits available, we can save a factor of $O(d)$ in the time complexity, as discussed in Appendix A.5.
>
> **For weakness 2:**
>
> The oracle assumptions (e.g., for function values and matrix access) are standard in the quantum optimization and quantum machine learning literature to isolate and quantify the potential for quantum speedups in principle. While challenging for current NISQ devices, research into physical implementations of such oracles is an active field.
>
> **For weakness 3:**
>
> We respectfully argue that there are still many technical challenges. For example, one of technical contributions lies in handling complex constraints like the Latent Group Norm, where we designed a quantum-parallel architecture for computing the dual norm of each group (Lemma 11) and, more critically, established the first error propagation analysis for dual norm computation under gradient approximation (Theorem 6). This provides a novel, quantum-specific technique to bound approximation errors and maintain convergence. The matrix domain also introduces innovations in iterative quantum multiplication error analysis (QPM) and streamlined top singular vector extraction (QTSVE).
>
> **For weakness 4:**
>
> The $\tilde{O}$ notation hides only polylog(d) factors, not arbitrary constants, and is used to focus on the primary scaling with dimension $d$ and precision $\varepsilon$, where we demonstrate a clear $\sqrt{d}$ to $d$ quantum advantage.
>
> **For question 1:**
>
> This is an excellent observation, and we thank the reviewer for making the connection to the specific solver in [1]. We will add a discussion in the revised manuscript about this BQP work [1]. We think our quantum techniques are directly applicable. As recognized in [1], solving BQP relies on computing top singular vectors—a task we have accelerated quantumly in our matrix-domain algorithms (Algorithms 3 & 4). Our QTSVE and QPM subroutines can thus serve as plug-in accelerators within classical BQP solvers.
>
> [1]. Li, S., et al., Tighter bound estimation for efficient biquadratic optimization over unit spheres, Journal of Global Optimization, 2024.
>
> **For question 2:**
>
> Our approach could provide a proven quantum speedup per iteration for the shared core subroutine (top singular vector computation). The convergence rate and final solution quality for BQP would be inherited from the host classical algorithm (e.g., alternating minimization). We accelerate the inner loop without altering the outer algorithm's theoretical guarantees.
>
>
> We thank the reviewer again for their time and insightful comments, and we hope that our responses have adequately addressed all points of concern. We will incorporate the above discussions and corresponding revisions into the revised manuscript.

---

> > ### Comment · Reviewer_CRBK · 2025-11-27
> >
> > I appreciate the authors’ response. However, I remain uncertain about the paper’s specific advantages, its practical research value, and how it differs from conventional algorithms.

---

> > > ### Author Response · Authors · 2025-11-27
> > >
> > > We thank the reviewer for their follow-up comment. We appreciate the opportunity to further clarify the core contributions of our work. Below, we directly address the three key points raised:
> > >
> > > **1 Specific Advantages**
> > >
> > > The primary and definitive advantage of our quantum algorithms is a provable reduction in query complexity. Technically, in the vector domain, we have essentially designed a more general Quantum Frank-Wolfe algorithm for atomic sets. The core acceleration still lies in locating the most dominant atom, but this can no longer be achieved by simply applying the finite-difference method as in the ℓ₁-norm case. The $\ell_1$-norm case—which is easier to understand and serves as a special instance of an atomic set—was placed in the main text primarily to facilitate comprehension of the overall framework. In the matrix domain, we essentially developed two complementary algorithms tailored to high-rank and low-rank gradient matrices, respectively. Furthermore, establishing a quantum advantage thorough end-to-end convergence analysis under the accumulated errors from all components—including our newly designed maximum dual norm estimation subroutine and power method subroutine, was nontrivial.
> > >
> > > **2 Practical Research Value**
> > >
> > > Our work provides a rigorous theoretical foundation and algorithmic blueprint for the upcoming era of hybrid quantum-classical optimization. As quantum hardware matures, our work precisely identifies which computational bottlenecks in projection-free optimization are amenable to quantum acceleration and provides the tools to exploit it. While heuristic algorithms and parameterized quantum circuit models may be more suitable for the NISQ-era hardware, quantum computing will not remain in the NISQ era forever. In fact, it is precisely the existence of algorithms with significant theoretical advantages—which require universal fault-tolerant quantum computers—that drives the continuous improvement of quantum hardware. Otherwise, if NISQ devices were sufficient, what would be the motivation for further advancing quantum computers?
> > >
> > > **3 Difference from Conventional Algorithms**
> > >
> > > Beyond the fundamental improvement in computational complexity, the most essential distinction between quantum and classical algorithms lies in their approach to two core components: the maximum atom search and the top singular vector extraction. Crucially, these are precisely the components that constitute the most computationally expensive bottlenecks in classical algorithms. The remainder of the framework still follows the classical Frank-Wolfe structure. We believe it is appropriate to focus quantum acceleration specifically on the classically challenging components, as quantum and classical computing each have their own strengths, and a hybrid approach that leverages both is both prudent and elegant.
> > >
> > > We sincerely appreciate the opportunity to further clarify the core contributions of our work. Should any concerns remain, would you mind kindly specifying them so we could address them more precisely?

---

### Official Review · Reviewer_jkkJ · 2025-10-31

**Soundness:** 2
**Presentation:** 2
**Contribution:** 2
**Rating:** 4
**Confidence:** 4

**Summary:**

The paper studies how quantum computing can accelerate projection-free optimization methods, particularly the Frank–Wolfe (FW) algorithm, which avoids costly projection steps by solving linear subproblems. For example, they studied for $\ell_1$-norm and simplex constraints for vector optimization problems, and nuclear norm constraints for matrix variables.

**Strengths:**

- Originality: the authors study quantum acceleration for projection-free convex optimization for several classes of optimization problems. They propose some ways to achieve quantum acceleration under the FW framework, like quantum gradient estimation, singular value extraction, et.c.

- Quality: They gave theoretical analysis of their methods, proving a worst case guarantee.

- Clarity: The writing of the paper is good. Their problem formulations, algorithms and theoretical results are presented in a structured way.

- Significance: Projection free methods for optimization is widely used and important in machine learning. Their results show the possibility of quantum speedups in this aspect.

**Weaknesses:**

The paper’s algorithmic advances rely heavily on existing quantum subroutines—such as quantum maximum finding, gradient estimation, and singular value extraction—raising questions about how much novelty lies beyond combining these tools within the Frank–Wolfe framework. The authors could better articulate what new technical challenges are overcome or what insights are unique to the projection-free setting. For example, what is the novelty of this work on vector variable problems compared with Chen & de Wolf (2023)?

Moreover, the analysis would benefit from a clearer discussion of applicability and limitations. The claimed speedups depend on assumptions like low-rank gradients and favorable spectral gaps; in dense or ill-conditioned cases, the advantage may vanish. Explicitly characterizing these regimes and their practical implications would strengthen the paper’s significance.

**Questions:**

- Clarification on Novelty and Technical Contributions: Beyond integrating known quantum primitives, what are the key new technical insights or analyses specific to the projection-free (Frank–Wolfe) setting? For example, are there difficulties in ensuring convergence or oracle compatibility that required new ideas?

- Regime of Quantum Advantage: The quantum speedups depend on parameters like the rank, singular value gap, and Lipschitz constants. Could you provide a more explicit characterization of the parameter regimes where your algorithms achieve practical advantage over classical FW? Also, are there examples (synthetic or theoretical) where these advantages disappear or become marginal?

- Clarification of the Matrix-Case Analysis: In the matrix domain, your algorithms assume precomputed gradients. Would including gradient computation change the overall asymptotic complexity or speedup factor? The analysis relies on rank and spectral gap assumptions—can you discuss robustness when these assumptions are only approximately satisfied?

---

> ### Author Response · Authors · 2025-11-20
>
> We sincerely thank the reviewers for their thoughtful comments and especially for their positive recognition of our work's originality, theoretical soundness, significance, and clarity of presentation. Below, we provide point-by-point responses to the reviewers' questions and concerns.
>
> **For question 1 and weakness 1:**
>
> In the vector case, our work is fundamentally distinct from Chen & de Wolf (2023) due to a core difference in the problem setting: we consider a general objective function accessible only through a function value oracle, which lacks a closed-form gradient. Consequently, the need for quantum gradient estimation and the subsequent handling of its inherent approximation errors are novel challenges unique to our work. For example, in the section of Latent Group Norm constraint, we encountered a fundamental difficulty as follows:
>
> 1.	The quantum parallelization of the constraint structure required a complete redesign of the computational paradigm. As detailed in Lemma 11, we devised a layered quantum-parallel architecture: creating a superposition over group indices, computing the gradients for all components of each group in parallel, and then parallelly calculating the dual norm for each group.
> 2.	We uncovered the challenge of error propagation in dual norm computation. In the classical setting, the dual norm can be calculated exactly. However, in the quantum setting, gradient estimation errors propagate through a complex transformation into the dual norm calculation. This is the core contribution of Theorem 6—we established a precise upper-bound analysis for the dual norm error. Leveraging Hölder's inequality, we proved that the error between the approximate dual norm and the true dual norm satisfies Inequality (83). This enables a carefully designed parameter schedule that ensures these approximation errors are absorbed into the standard convergence rate of FW.
>
> Furthermore, the $\ell_1$-norm constrained case is merely a special instance of the latent group constraints. We chose to present the $\ell_1$-norm case in the main text due to its readability, which facilitates understanding of the core concepts. The technical novelty in the matrix section include the error analysis for iterative quantum matrix multiplication in QPM, the streamlining of the Bellante et al. (2022) method via quantum maximum finding in QTSVE, and the analysis and adaptation of quantum maximum finding for non-uniform superposition states in QTSVE.
>
> **For question 2 and weakness 2:**
>
> The quantum advantage in the vector case is unconditional, while the main focus of our discussion regarding parameter-dependent advantage lies in the matrix domain, as elaborated in Appendix A.5.
> For Algorithm 3, quantum advantage exists when $d >r / \sqrt{\sigma_1 - \sigma_2}  \epsilon$. For Algorithm 4, quantum advantage holds when $d > \sqrt{r} \sqrt{\sigma_1 - \sigma_2} /  \epsilon^2  (1 - \sigma_1)$. Since the quantum subroutines in the matrix section effectively process the gradient matrix normalized by its Frobenius norm, when this matrix has very low rank, $1 - \sigma_1$ tends to be small (approaching 0 when the rank is 1). In such cases, Algorithm 3 delivers better performance, whereas Algorithm 4 is more suitable otherwise. We will incorporate this discussion into Appendix A.5.
>
> **For question 3:**
>
> While estimating the gradient for a matrix-variable function with a zeroth-order oracle requires only O(1) queries, the process encounters the well-known I/O bottleneck in quantum linear system algorithms. The additional overhead lies in reading this gradient into quantum memory, as discussed in Remark 3. This challenge remains a significant open problem in the quantum community. The current convention, which we follow, is to assume the gradient is pre-loaded to analyze the core computational speedup. As noted in Remark 3, currently, this bottleneck can be mitigated for sparse gradient matrices, where the loading cost is reduced. We await further advancements from the quantum community to address this general I/O issue.
>
> Our analysis explicitly incorporates rank and spectral gap, and the existence of two complementary algorithms (QTSVE and QPM) provides robustness. When the matrix is strictly low-rank with a clear spectral gap, Algorithm 3 is optimal. For higher-rank matrices, Algorithm 4 with its $\sqrt{r}$ dependence offers a more robust alternative. This algorithmic choice itself constitutes a robust strategy across different spectral regimes.
>
> We sincerely thank the reviewers for their insightful feedback and valuable suggestions. The points raised have helped us significantly improve the clarity and depth of our work. We will incorporate all the above discussions and corresponding revisions into the revised manuscript. We hope that our responses have adequately addressed all points of concern.

---

### Official Review · Reviewer_26XM · 2025-10-31

**Soundness:** 2
**Presentation:** 3
**Contribution:** 2
**Rating:** 4
**Confidence:** 3

**Summary:**

This paper investigates projection-free optimization for sparse convex problems. It proposes two quantum algorithms—handling vector and matrix domains respectively—that achieve improved complexities by reducing the dependence on the dimension d.

**Strengths:**

The paper is well-written, featuring a clear presentation of the methods, the obtained complexity results, and comparisons with prior works.

**Weaknesses:**

My primary concern pertains to the paper's motivation. I find the core motivation insufficiently justified. The manuscript does not adequately establish a compelling need for specialized quantum algorithms for this particular class of problems. From my perspective, the proposed method appears somewhat ad-hoc, lacking both a clear demonstration of practical application and the provision of new, general insights for the field of quantum algorithm design.

**Questions:**

1.	What is the motiveitaion of this paper? Why do we need to solve this type of problem?
2.	Why do we need to accelerate Frank-Wolfe algorithms? Is it possible to design special algorithm for these sparse learning problems?
3.	Recent advanves in non-convex optimization often solves sparse (low-rank) problem by reformulating the problem by a quadatratic parametrization problem? Why do you work on this way?

---

> ### Author Response · Authors · 2025-11-20
>
> We thank the reviewer for the thoughtful comments. We are very pleased that the reviewer finds our paper well-written and acknowledges our clear presentation of the methods, the detailed complexity results, and the thorough comparisons with prior works. Below we take this opportunity to clarify the motivation behind our work by addressing your questions point-by-point.
>
> **For question 1:**
>
> The optimization over structured sparsity-inducing constraint sets like the $\ell_1$-ball, simplex and the nuclear norm ball is a cornerstone of modern sparsity and low-rank learning. This framework is not an academic niche but the engine behind a vast range of critical applications, including sparse regression (Lasso), sparse signal recovery, matrix completion, boosting algorithms (e.g., AdaBoost), Support Vector Machines, and density estimation [Jaggi13]. Also thank Reviewer jxNP for providing more applications: signal processing (sparsity constraints via l1 norm), game theory (zero sum games with simplex) and SDPs (nuclear norm optimization).
>
> [Jaggi13] Jaggi, Martin. "Revisiting Frank-Wolfe: Projection-free sparse convex optimization." International conference on machine learning. PMLR, 2013.
>
> **For question 2:**
>
> The Frank-Wolfe algorithm is the premier method for such problems due to its projection-free property. However, its scalability is fundamentally limited by the cost of the Linear Minimization Oracle (LMO), which scales linearly with the problem dimension $d$. Prior work by [Chen23] provided a preliminary quantum acceleration for a very specific case. Our work, in contrast, conducts the first systematic investigation of quantum-accelerated FW for the general setting—using only a function value oracle and handling a wider range of constraints. We thereby establish a theoretical benchmark for quantum advantage across this entire, application-rich domain.
>
> [Chen23] Chen, Yanlin, and Ronald de Wolf. "Quantum Algorithms and Lower Bounds for Linear Regression with Norm Constraints." 50th International Colloquium on Automata, Languages, and Programming (ICALP 2023).
>
> **For question 3:**
>
> When you mentioned "reformulating the problem by a quadratic parametrization problem," were you referring to the Burer-Monteiro approach? Methods like Burer-Monteiro are powerful but specialized, often relying on stronger structural assumptions (e.g., for SDPs) for their theoretical guarantees. In contrast, the convex FW framework provides global convergence guarantees for the entire problem class. It works directly on the original constraint sets without needing problem-specific reformulations. This makes our quantum acceleration results broadly applicable and theoretically robust, establishing a foundational benchmark for the field. Thank you for raising this point. Designing quantum algorithms for problems with specific structures using the Burer-Monteiro framework represents a promising direction for future research.
>
> Thank you again for your thoughtful comments, which have strengthened our paper. We hope that our responses have adequately addressed all points of concern.

---

### Official Review · Reviewer_xYdV · 2025-10-31

**Soundness:** 3
**Presentation:** 3
**Contribution:** 2
**Rating:** 2
**Confidence:** 3

**Summary:**

The paper is focused on convex optimization with convex constraints in high dimensional settings. The authors propose a quantum approach for achieving quadratic speedup, in terms of the problem dimensionality, by using quantum computing to solve the subproblems in Frank-Wolfe optimization algorithm.

**Strengths:**

The paper provides an original approach towards solving constraint optimization problems. The significance of the result is diminished, as noted in weaknesses, but deficiencies in complexity analysis of the method and its assumptions. The paper is clearly written, although the assumptions should be more prominently analysed.

**Weaknesses:**

The main weakness involves unrealistic assumptions that affect the complexity of the full algorithm. For example, step 7 in Alg. 2 involves preparing, in each Frank-Wolfe iteration, a state
$\sum_{i=0}^{d-1}|i>|x^{(t)}>|0>$, with $|x>$ defined in Assumption 3 to be $|x> = |x_1>|x_2>…|x_d>$. The state is over $d+2$ qubits. Similar qubit $O(d)$ qubit requirement is present in the rest of scenarios considered in the paper. This linear qubit requirement severely limits the applicability of the method for the stated goal of addressing high dimensional (high $d$) optimization problems.

Preparing a $d$-dimensional arbitrary qubit state is hard in general, yet the impact of this step on the complexity of the whole method is not discussed.

Some terminology could be revised, e.g. authors mention “sparsity-constrained problems” in the introduction, but actually address L1 norm and not L0 norm constraints.

**Questions:**

Are there any specific properties of $|x^{(t)}>$ over the iterations that would make state preparation efficient?

---

> ### Author Response · Authors · 2025-11-20
>
> We thank the reviewer for the thoughtful feedback and for acknowledging the originality and clarity of our work. We appreciate the opportunity to address the concerns raised, which we believe will help improve the manuscript. Below, we provide a point-by-point response.
>
> **Response to the Concern on "Preparing a d-dimensional arbitrary qubit state is hard in general":**
>
> The state $|x^{(t)}⟩$ defined in Assumption 3 is not an arbitrary d-dimensional quantum state that requires amplitude encoding. Rather, it is a computational basis state where the classical vector $x^{(t)}$ is stored in binary representation across $O(d)$ qubits. The preparation of such a state is remarkably simple: It can be achieved with a circuit of depth $1$ using at most $O(d)$ single-qubit gates (specifically, Pauli-X gates). a cost directly analogous to the $O(d)$ memory operations required in the classical algorithm.
>
> Furthermore, we thank the reviewer for the inspiration to elaborate on a more refined analysis. Indeed, the specific structure of the Frank-Wolfe algorithm under sparsity-inducing constraints (like the $\ell_1$-norm) allows for an even more efficient state preparation process than the general case. A key insight is that if we initialize the algorithm at $x^{(0)} = 0$, each Frank-Wolfe step adds a single coordinate direction to the solution. Specifically, the update rule $x^{(t+1)} = (1-γ_t)x^{(t)} + γ_t s_t$—where $s_t$ is a standard basis vector—implies that the solution $x^{(t)}$ after $t$ iterations is a sparse vector with at most $t$ non-zero components. Consequently, the quantum state $|x^{(t)}⟩$ is a sparse computational basis state. Preparing this state does not require initializing all $d$ components from scratch in every iteration. Instead, we can perform an incremental update, setting at most one new coordinate to a non-zero value per iteration. The gate complexity for this sparse update is $O(t)$. Crucially, the total number of iterations $T$ required for an $ε$-optimal solution is $T = O(1/ε)$, which is independent of the dimension $d$. Therefore, the state preparation overhead per iteration remains $O(1/ε)$, completely decoupled from the potentially large dimension $d$.
>
> In summary, the preparation of the quantum states in our algorithm is not a prohibitive bottleneck. The initial state is trivial, and subsequent states are sparse and efficient to update. We will incorporate this clarifying discussion into the revised manuscript.
>
> **Regarding the terminology of "sparsity-constrained problems":**
>
> The term "sparsity-constrained problems" is sometimes used as a shorthand for problems employing sparsity-promoting convex relaxations, most notably the L1-norm, which is the convex envelope of the L0-norm. To eliminate any potential ambiguity and enhance precision, we will replace "sparsity-constrained problems" with the more accurate description "sparsity-inducing constrained problems" throughout the revised manuscript.
>
> We thank the reviewer again for their constructive comments. We believe that with the clarifications and revisions outlined above, the manuscript will be strengthened. If have any further question/concern, feel free to discuss!

---

### Author Response · Authors · 2025-11-28

As we have not received any further questions or concerns following our response, we have now optimized our manuscript based on the existing discussions and uploaded the revised version, with new additions highlighted in blue. Following the reviewer's suggestions, the revisions primarily focus on writing improvements, including more comprehensive discussions of motivations, technical contributions, potential applications, and the complementary relationship between the two algorithms in the matrix section. Technically, we supplemented the quantum resource costs for Theorem 6, and for Assumption 3, we have added a dimension-independent fast preparation method of input states in the projection-free sparse convex optimization scenario.

We once again thank the reviewers for their insightful comments and time spent on improving the quality of our work, and we look forward to receiving further feedback and a re-evaluation of the scores.

---

### Meta-Review · Area_Chair_7EpZ · 2025-12-24

**Summary:**

This paper studied quantum algorithms for projection-free sparse convex optimization. In particular, it proposed a quantum Frank-Wolfe method with detailed complexity bounds.

There are 5 reviews, 4 of which are negative and 1 is positive. It is appreciated that the results are technically solid, but it applied mostly existing quantum subroutines (and hence the technical contribution might not reach the bar of ICLR), and it does not have discussion about practical applications - not even complexity bounds for explicit sparse convex optimization problems such as the matrix completion problem mentioned in Eq. (2). Considering both these points as well as the overall negative scores, the decision is rejection.

**Reviewer Concerns:**

The authors addressed in the rebuttal about the settings as well as relationships to previous literature. However, technical novelty and practical impacts (this paper is not only pure theoretical without experiments but only studied convex optimization in an oracle setting, not practical optimization problems) are essential limits outstanding here.

**Reviewer Scores:**

There might be minor score changes, but I don't think it will overturn the negative scores overall.

---

### Decision · Program_Chairs · 2026-01-26

Reject